# Twinfilin modulates tissue contractility through uncapping of capping protein in *C. elegans*

Anupreet Saini[1], Shir Kreizman[1], Ekram Towsif[2], Jonathan Martinez-Lopez[2], Iska Maimon Zielonka[1], Anat Nitzan[1], Lee Rudnick[1], Shashank Shekhar[2] and Ronen Zaidel-Bar[1,*]

## ABSTRACT

The actin cytoskeleton is dynamically remodeled by conserved regulators to control cell and tissue mechanics. Although many of these proteins are well studied, their roles in driving tissue-specific contractility remains unclear. Twinfilin, an actin uncapper and depolymerase, has not previously been linked to tissue contractility. Here, we show that the sole twinfilin ortholog in *C. elegans*, TWF-2, modulates actomyosin contractility in the spermatheca. TWF-2 localizes to the spermathecal cortex through interactions with α-spectrin (SPC-1) and β-spectrin (UNC-70). *In vitro*, TWF-2 inhibits depolymerization of ADP-actin filaments and modestly promotes depolymerization of ADP-$P_i$-actin filaments, while also rapidly displacing capping protein (CAP-1 and CAP-2) from filament barbed ends. *In vivo*, loss of TWF-2 alone does not produce an obvious phenotype, but simultaneous loss of TWF-2 partially rescues the embryonic lethality caused by CAP-1 depletion. Likewise, loss of the contractility regulator SPV-1 results in elevated F-actin and phosphorylated myosin, leading to hypercontractility; this phenotype is suppressed by removing TWF-2, which lowers F-actin levels without affecting myosin or its phosphorylation. These findings demonstrate that TWF-2 modulates actin dynamics in a tissue-specific manner. This work provides the first *in vivo* evidence that twinfilin modulates contractility and reveals how its interactions with capping protein and spectrins help maintain balanced actomyosin levels in the spermatheca.

KEY WORDS: Twinfilin, Actin dynamics, Actin capping protein, Actomyosin contractility, *C. elegans*, Spermatheca, Tissue-specific regulation, Spectrin

## INTRODUCTION

The actin cytoskeleton is a dynamic filamentous network essential for fundamental cellular processes such as cytokinesis, motility and cell shape changes. Actin-binding proteins – including those that interact with actin monomers or specific filament ends – play central roles in shaping actin architecture and dynamics, ensuring these processes occur at the right time and place. Among these regulators, twinfilin is an evolutionary conserved key modulator of actin dynamics, with diverse functions across species (Ulrichs and Shekhar, 2025).

In yeast, deletion of twinfilin causes morphological defects in budding cells (Goode et al., 1998). In *Drosophila*, twinfilin regulates bristle formation (Wahlström et al., 2001), border cell migration and axonal outgrowth (Wang et al., 2010). In mammals, although initially reported as dispensable for mouse development (Nevalainen et al., 2011), twinfilin is essential for platelet function in mice (Stritt et al., 2017) and for neurite growth and spine density in cultured rat neurons (Wang et al., 2021; Yamada et al., 2007). In humans, both reduced and elevated twinfilin expression have been implicated in cancer progression (Bockhorn et al., 2013; Samaeekia et al., 2017; Zhai et al., 2023).

Despite these species-specific phenotypes, twinfilin consistently acts on actin to modulate its dynamics. Initially identified as an actin monomer-binding protein (Goode et al., 1998; Palmgren et al., 2001; Vartiainen et al., 2003), twinfilin is now known to interact with filament barbed ends, in a context dependent manner. It inhibits barbed-end depolymerization of ADP–actin filaments (Hakala et al., 2021) but accelerates depolymerization of ADP-$P_i$-actin filaments (Shekhar et al., 2021). Importantly, twinfilin-mediated barbed-end depolymerization persists even under assembly-promoting conditions in the presence of profilin-ATP-G-actin (Hakala et al., 2021; Shekhar et al., 2021). A central feature of twinfilin function is its interplay with actin capping protein (CP), a major barbed end binding protein that terminates filament elongation (Falck et al., 2004; Palmgren et al., 2001). Twinfilin binds CP directly through a C-terminal CP-interacting (CPI) motif. Although initially thought to support CP activity by protecting it from CARMIL inhibition (Johnston et al., 2018), later work revealed that twinfilin can act as an uncapper, displacing CP from barbed ends (Hakala et al., 2021; Mwangangi et al., 2021; Reddy et al., 2025; Ulrichs et al., 2023). Interestingly, while mammalian twinfilin alone is a relatively weak uncapper and only accelerates uncapping sixfold (Hakala et al., 2021), it synergizes with V-1/myotrophin and formin Diaph1 to accelerate uncapping by over 50-fold and 300-fold, respectively (Hakala et al., 2021; Reddy et al., 2025). These findings highlight the complex, context-dependent regulation of actin dynamics by twinfilin (Ulrichs and Shekhar, 2025). However, while the biochemical interplay between twinfilin and CP is well characterized *in vitro*, its functional consequences in tissues remain poorly understood.

To investigate the *in vivo* role of these conserved actin regulators, we turned to *Caenorhabditis elegans*, focusing on its reproductive system, which consists of two U-shaped gonads. Each gonad comprises somatic tissues housing germ cells that differentiate into sperm and oocytes, with sperm stored in a specialized contractile fertilization chamber, the spermatheca. CP is broadly expressed in the *C. elegans* reproductive system, including the germline and

[1]Gray School of Medical Sciences, Gray Faculty of Medical and Health Sciences, Tel Aviv University, P.O.B 39040, Ramat Aviv, Tel Aviv 69978, Israel. [2]Departments of Physics, Cell Biology and Biochemistry, Emory University School of Medicine, 400 Whitehead Biomedical Research Building, 615 Michael Street, Atlanta, GA 30322, USA.

*Author for correspondence (zaidelbar@tauex.tau.ac.il)

A.S., 0000-0002-7466-311X; S.K., 0009-0002-1949-6512; S.S., 0000-0003-1494-0151; R.Z., 0000-0002-1374-5007

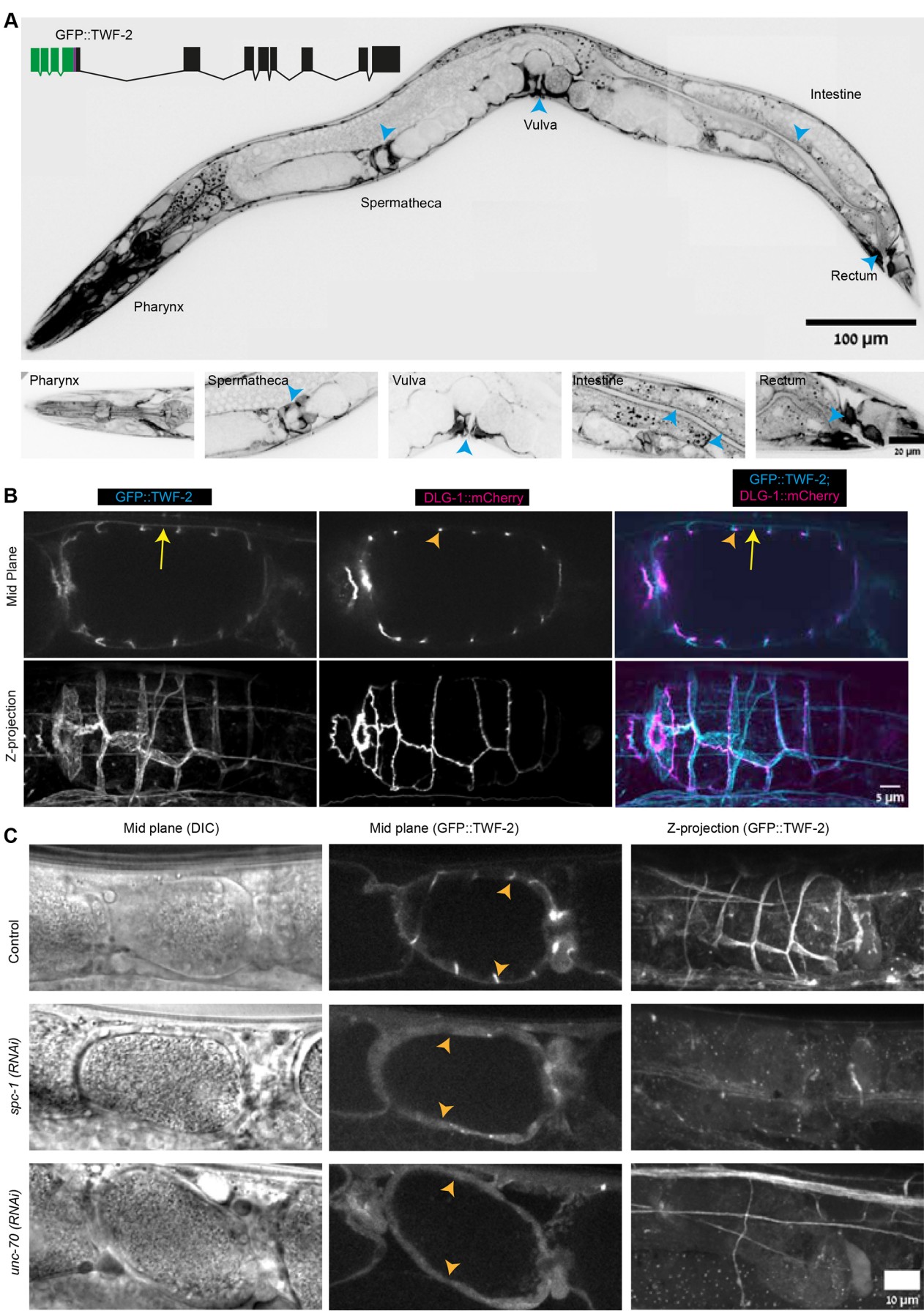

**Fig. 1.** See next page for legend.

**Fig. 1. Localization of endogenous TWF-2 in adult *C. elegans* contractile tissues.** (A) Expression pattern of endogenous TWF-2 tagged with GFPnovo (henceforth GFP) in adult hermaphrodites. Top: Representative mid plane showing GFP::TWF-2 expression in contractile tissues (pharynx, spermatheca, vulva, intestine and rectum; blue arrowheads). Scale bar: 100 µm. Schematic inset: Representation of the fusion of GFP (green) and TWF-2 (black) with exons (full boxes), introns (thin lines) and linker (purple). Bottom: Magnified views of indicated regions are shown. Scale bar: 20 µm. (B) Subcellular localization of GFP::TWF-2 (cyan) relative to the apical marker DLG-1::mCherry (magenta) in spermathecal cells. Single mid-plane (upper panel) and maximum z-projection (lower panel) views demonstrate basolateral TWF-2 localization (yellow arrows) distinct from apical DLG-1 (orange arrowheads). Scale bar: 5 µm. (C) RNAi-mediated depletion of spectrins disrupts TWF-2 cortical localization. Representative images of spermathecae from worms treated with control, *spc-1* and *unc-70* RNAi. Cortical GFP::TWF-2 signal (orange arrowheads) is disrupted upon spectrin knockdown. *N*=2 biological replicates with *n*≥40 worms analyzed per condition. Scale bar: 10 µm.

spermatheca. Its role in germline architecture is established, as CP depletion causes a hypercontractile germline marked by elevated F-actin and myosin (Ray et al., 2023). The spermatheca, which contains a highly organized actomyosin network that drives ~150 rounds of ovulation, provides an ideal model to dissect how actin regulators control tissue contractility. Spermathecal contractility depends on multiple regulators, including the RhoGAP spermatheca physiology variant-1 (SPV-1), a mechanosensor that suppresses contractility by inactivating RHO-1 (Tan and Zaidel-Bar, 2015), and spectrins, which support contractility and preserve actin network integrity (Wirshing and Cram, 2018). Notably, no link between twinfilin and spectrins has been reported, and a role for twinfilin in regulating actomyosin contractility has not been described.

In this study, we characterize the expression, localization and function of the only twinfilin in *C. elegans*, TWF-2, an ortholog of human TWF2. Specifically, we investigate how TWF-2 interacts with CP and spectrins, and how it modulates actomyosin contractility in the spermatheca. We demonstrate that TWF-2 is broadly expressed in the worm and localizes to the spermathecal cortex in a spectrin-dependent manner. Loss of TWF-2 alleviates hypercontractility caused by CP and SPV-1 dysfunction. *In vitro* assays confirm a direct association of *C. elegans* CP and TWF-2, providing mechanistic insight into their functional relationship. Together, these findings establish a role for twinfilin in tissue contractility and establish a framework for understanding its contributions to other contractile tissues across species.

## RESULTS
### TWF-2 is widely expressed in contractile tissues and its cortical localization in the spermatheca is dependent on spectrins

To examine the expression pattern and subcellular localization of TWF-2 in adult *C. elegans* hermaphrodites, we tagged endogenous TWF-2 with the fluorescent marker GFPnovo2 (henceforth referred to as GFP) at its N-terminus (Fig. 1A) using CRISPR/Cas9-mediated genome editing. To verify that tagging TWF-2 did not disrupt development, we measured the brood size and embryonic lethality of the GFP::TWF-2 strain and found them to be comparable to wild type (Fig. S1A,B).

Imaging of GFP::TWF-2 revealed widespread expression across multiple tissues, with particularly prominent expression in the pharynx, spermatheca, vulva, intestine and rectum, which are all contractile tissues (Fig. 1A). This spatial distribution suggested a potential role of TWF-2 in regulating non-muscle actomyosin contractility in *C. elegans*.

Next, we examined its subcellular localization and chose to focus on the spermatheca. Within spermathecal cells, we observed TWF-2 to be concentrated near the plasma membrane, in what appeared to be cortical localization. To validate its localization at the cellular cortex we crossed TWF-2::GFP with DLG-1::mCherry, a known apical junction protein, and observed that TWF-2 mainly localized to the basolateral membranes of spermathecae (Fig. 1B). To identify proteins that interact with TWF-2 and possibly regulate its subcellular localization, we performed co-immunoprecipitation (co-IP) followed by mass spectrometry (MS) using the GFP::TWF-2 strain, with N2 as a control (Fig. S1C). Compared to the N2 control, TWF-2 was enriched ~2000-fold in the GFP::TWF-2 sample, confirming the pull down assay. Based on enrichment scores and the known role of TWF-2 as an actin-interacting protein, we shortlisted cytoskeletal candidates for further investigation (Table S1A). These candidates were depleted by RNAi followed by evaluation of TWF-2 localization in the GFP::TWF-2 strain.

From the candidate genes screened, RNAi knockdown of *spc-1* (α-spectrin) and *unc-70* (β-spectrin) disrupted GFP::TWF-2 cortical localization in the spermatheca (Fig. 1C). Spectrin knockdown also disrupted TWF-2 cortical localization in the vulva, but not in other contractile tissues where TWF-2 is expressed (Fig. S2A), suggesting a tissue-specific role of spectrins in the cortical localization of TWF-2. Notably, *twf-2* RNAi, which knocked down 76% of TWF-2, did not affect the localization of SPC-1 (Fig. S2B,C), suggesting that TWF-2 is not required for spectrin localization. Generation of a *twf-2* null mutant (described in the next section) allowed us to confirm this: crossing SPC-1::mKate with the *twf-2* null did not change the SPC-1 distribution (Fig. S2E). To check if SPC-1 and TWF-2 co-localize, we crossed SPC-1::mKate strain with GFP::TWF-2. Quantitative analysis revealed strong co-localization between them, with a Pearson correlation coefficient of 0.9730 (Fig. S2F,G), further supporting their functional association.

### TWF-2 loss of function partially rescues the phenotype of CP knockdown and the two proteins co-localize in some tissues

In other model systems, loss of twinfilin leads to structural and functional defects. To elucidate the function of TWF-2 in *C. elegans* we generated a *twf-2* deletion strain using CRISPR/Cas9 to cut out most of the endogenous locus (Fig. S2D). Surprisingly, the TWF-2 deletion mutant exhibited no discernible phenotype under standard conditions. To evaluate a possible effect on the level or organization of F-actin in the *twf-2* deletion strain, we crossed it with a strain expressing the F-actin-bundling protein plastin/PLST-1 endogenously tagged with GFP. Examination of PLST-1::GFP in embryos did not reveal any difference between the *twf-2* KO and control PLST-1::GFP worms (Fig. S3A).

We hypothesized that the lack of phenotype in *twf-2* null worms might reflect redundancy with other regulators of actin dynamics, such that its role would only become evident when another regulator was depleted. To test this, we compiled a list of actin regulators and knocked them down individually by RNAi in the *twf-2* null background (Table S1B). We then compared the resulting embryonic lethality to that observed with the same RNAi in wild-type worms. Unexpectedly, no gene knockdown produced higher embryonic lethality in the *twf-2* null strain as compared to controls. In contrast, depletion of the CP subunit CAP-1 produced a milder phenotype in the absence of *twf-2*. Specifically, *cap-1* RNAi led to 47±19% (mean±s.d.) embryonic lethality in wild-type worms but only 28±10% embryonic lethality in *twf-2* null worms (Fig. 2A).

Previous studies in other model systems and *in vitro* experiments have demonstrated that twinfilin and CP often closely associate to

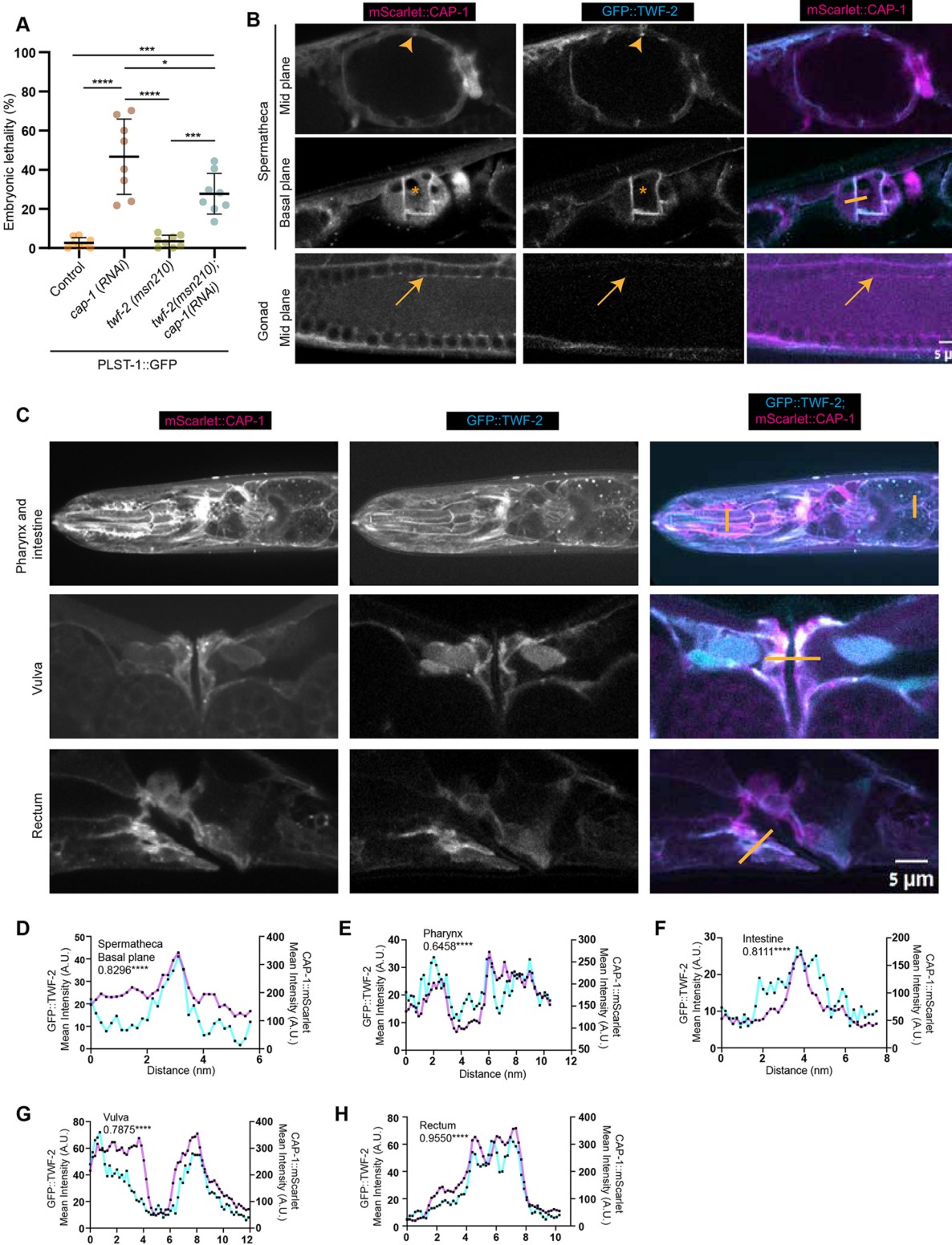

**Fig. 2. Genetic interaction between TWF-2 and CAP-1 and their tissue-specific colocalization.** (A) Embryonic lethality assay. Dot plots show the percentage of embryonic lethality for control (wild type), *cap-1 RNAi* on wild type, *twf-2(msn210)* mutant control and *twf-2(msn210); cap-1(RNAi)*. Statistical significance was determined by ordinary one way ANOVA (*P<0.05, ***P<0.001, ****P<0.0001). N=3 biological replicates with n≥298 embryos analyzed for each condition. Data are mean±s.d. (B) Subcellular co-localization of endogenous GFP::TWF-2 (cyan) and mScarlet::CAP-1 (magenta) in reproductive tissues. Cortical co-localization in spermatheca (arrowheads) contrasts with CAP-1-only regions in spermathecal cytoplasm (asterisks) and germline rachis (arrows). Scale bar: 5 µm. (C) Co-localization of endogenous GFP::TWF-2 (cyan) and mScarlet::CAP-1 (magenta) in other contractile tissues (pharynx, intestine, vulva and rectum). Orange lines indicate regions used for line profile quantification in D-H. Scale bar: 5 µm. (D-H) Line profile analyses of fluorescence intensity for GFP::TWF-2 (cyan) and mScarlet::CAP-1 (magenta) across indicated regions in B and C. Pearson correlation coefficients are indicated at top left of each graph.

regulate actin dynamics (Hakala et al., 2021). To investigate whether this association also occurs in *C. elegans*, we crossed the endogenously-tagged mScarlet::CAP-1 strain with the GFP::TWF-2 strain and examined their relative localization. We observed substantial co-localization of CAP-1 and TWF-2 in the spermathecal cortex, pharynx, intestine, vulva and rectum, suggesting a potential functional interaction in these tissues (Fig. 2B-H). However, we also observed the localization of CP in certain tissues where TWF-2 was absent. For example, CAP-1 prominently localized to the germline, while TWF-2 was not detected there (Fig. 2B). Moreover, also in tissues where both proteins were expressed, they did not always co-localize. For example, CAP-1 was present in the cytoplasm of the spermatheca while TWF-2 was notably absent from this region (Fig. 2B). Also, with respect to embryos, CAP-1 was expressed from early stages while TWF-2 expression started only during late embryogenesis (Fig. S3B). These findings indicate that CP and TWF-2 are not always required together but instead might participate jointly in some processes and separately in other processes, depending on the context. Furthermore, it appears that TWF-2 functions as a negative regulator of CP activity, since its loss alleviated the consequences of *cap-1* RNAi.

### TWF-2 promotes barbed-end depolymerization and is a potent uncapper of CP from actin filaments

Based on the previous results, we hypothesized that in *C. elegans*, twinfilin is an uncapper of CP. To test this and biochemically characterize the activity of TWF-2, we performed a series of *in vitro* experiments using microfluidics-assisted total internal reflection fluorescence (mf-TIRF) microscopy (Shekhar, 2017). *C. elegans* TWF-2 and CP (CAP-1 and CAP-2) were purified as recombinant proteins from *Escherichia coli*. Building upon previous work demonstrating the actin-depolymerizing activity of twinfilin (Shekhar et al., 2021), we first examined its effects on free barbed ends. Actin filaments with their barbed ends free were elongated from coverslip-anchored spectrin-actin seeds (Fig. 3A). The filaments were then aged for 15 min to generate ADP-actin filaments. Consistent with previous studies on mammalian twinfilin, we observed that 5 µM TWF-2 significantly decreased the depolymerization rate of aged ADP-actin filaments compared to buffer controls (Fig. 3B). Notably, this activity was specific to ADP-actin, as TWF-2 showed an opposite effect by increasing the rate of barbed-end depolymerization of unaged ADP-P$_i$-actin filaments (Fig. 3C), suggesting nucleotide-state-dependent regulation.

We next investigated whether TWF-2 could displace CP from filament barbed ends (Fig. 3D). By anchoring biotinylated SNAP-tagged *C. elegans* heterodimeric CP to coverslips and capturing preformed Alexa-488-labeled actin filaments, we visualized uncapping events in real-time. Time-lapse imaging revealed that TWF-2 induced rapid filament detachment (Fig. 3E,F), with 5 µM TWF-2 uncapping ∼80% of filaments within 100 s (Fig. 3F,G). Quantitative analysis demonstrated a concentration-dependent increase in uncapping rates by TWF-2, eventually reaching a saturation level (Fig. 3H). These results provide direct biochemical evidence that *C. elegans* TWF-2 functions both as a barbed-end depolymerizer and as a potent uncapping factor, thereby negating the role of CP.

Hence, we hypothesized that the partial rescue of *cap-1* RNAi-induced embryonic lethality in *twf-2* null worms occurs because, in the absence of the uncapper TWF-2, the remaining CP retains greater capping activity. An alternative explanation is that the rescue is indirect, reflecting loss of a twinfilin function independent of CP.

To distinguish between these possibilities, we asked whether loss of *twf-2* could rescue a complete absence of *cap-1*. We therefore crossed the *twf-2* null strain with a *cap-1* null mutant. Because homozygous *cap-1* null animals are inviable (100% larval lethal), the strain is maintained as a balanced heterozygous. Analysis of progeny from *cap-1/+;twf-2* mothers revealed that homozygous *cap-1;twf-2* double mutants all arrested as larvae, identical to *cap-1* null mutants alone, with no phenotypic rescue (Fig. 3I). Thus, a minimum level of CP is required for *twf-2* loss to reduce lethality, supporting the conclusion that TWF-2 functions as a CP uncapper in *C. elegans*.

### Loss of *twf-2* alleviates *cap-1* RNAi-associated embryonic lethality via F-actin modification in the spermatheca actin stress fibers

Previously, we have shown that *cap-1* RNAi in the germline causes a twofold increase in F-actin and myosin, leading to a hypercontractile phenotype (Ray et al., 2023). However, TWF-2 is not expressed in the germline (Fig. 2B). Thus, the partial rescue of *cap-1* RNAi-induced embryonic lethality in *twf-2* null worms is unlikely to arise from interactions in this tissue. In contrast, we found both CAP-1 and TWF-2 to be highly expressed and co-localized in the spermatheca (Fig. 2B). Since spermathecal hypercontractility is known to disrupt embryonic morphology – through excessive squeezing and occasional pinching of embryos – resulting in embryonic lethality (Tan and Zaidel-Bar, 2015), we hypothesized that loss of *twf-2* might partially rescue *cap-1* RNAi lethality by suppressing spermathecal hypercontractility.

To test this, we performed *cap-1* RNAi in wild-type and *twf-2* worms, and quantified the embryonic morphology. Correlation plots of embryo area and axial ratio revealed pronounced morphological defects in *cap-1* RNAi embryos, consistent with spermathecal dysfunction. Strikingly, deletion of *twf-2* partially rescued these defects, reducing the frequency of embryos with extreme morphologies (Fig. 4A-E). These findings suggest that TWF-2 modulates *cap-1* RNAi-associated lethality through its role in spermathecal contractility.

To directly assess whether *twf-2* loss modifies spermathecal contractility via effects on F-actin, we performed Phalloidin staining on extruded spermatheca from wild-type (N2) and *twf-2* mutant under control and *cap-1* (RNAi) conditions. As expected, *cap-1* RNAi in N2 worms caused a significant increase in F-actin levels in the spermatheca. Remarkably, this increase was completely suppressed in the *twf-2* mutant (Fig. 4F,G). Together, these findings confirm that TWF-2 functions as an uncapper in the spermatheca, mitigating the embryonic lethality associated with *cap-1* loss.

### Loss of *twf-2* alleviates spermathecal contractility in *spv-1* mutants

Our data suggest that TWF-2 modulates spermathecal actomyosin contractility through its uncapping of CP activity. However, since CP is also expressed in other tissues (Fig. 2) and in embryos (Fig. S3B), the tissue specificity of our observations remains unclear. To directly probe spermatheca-specific effects, we turned to a model of increased contractility restricted to this tissue. Previously, we have shown that loss of the RhoGAP SPV-1, which is expressed exclusively in the spermatheca, leads to ∼40% embryonic lethality due to hypercontractility driven by excess RHO-1 activity (Tan and Zaidel-Bar, 2015).

Strikingly, systemic *spv-1* RNAi in *twf-2* null mutants reduced embryonic lethality by 63% (19.2±8% in control versus 7±5% in *twf-2* null; Fig. 5A). Similarly, systemic *twf-2* RNAi in *spv-1* mutants

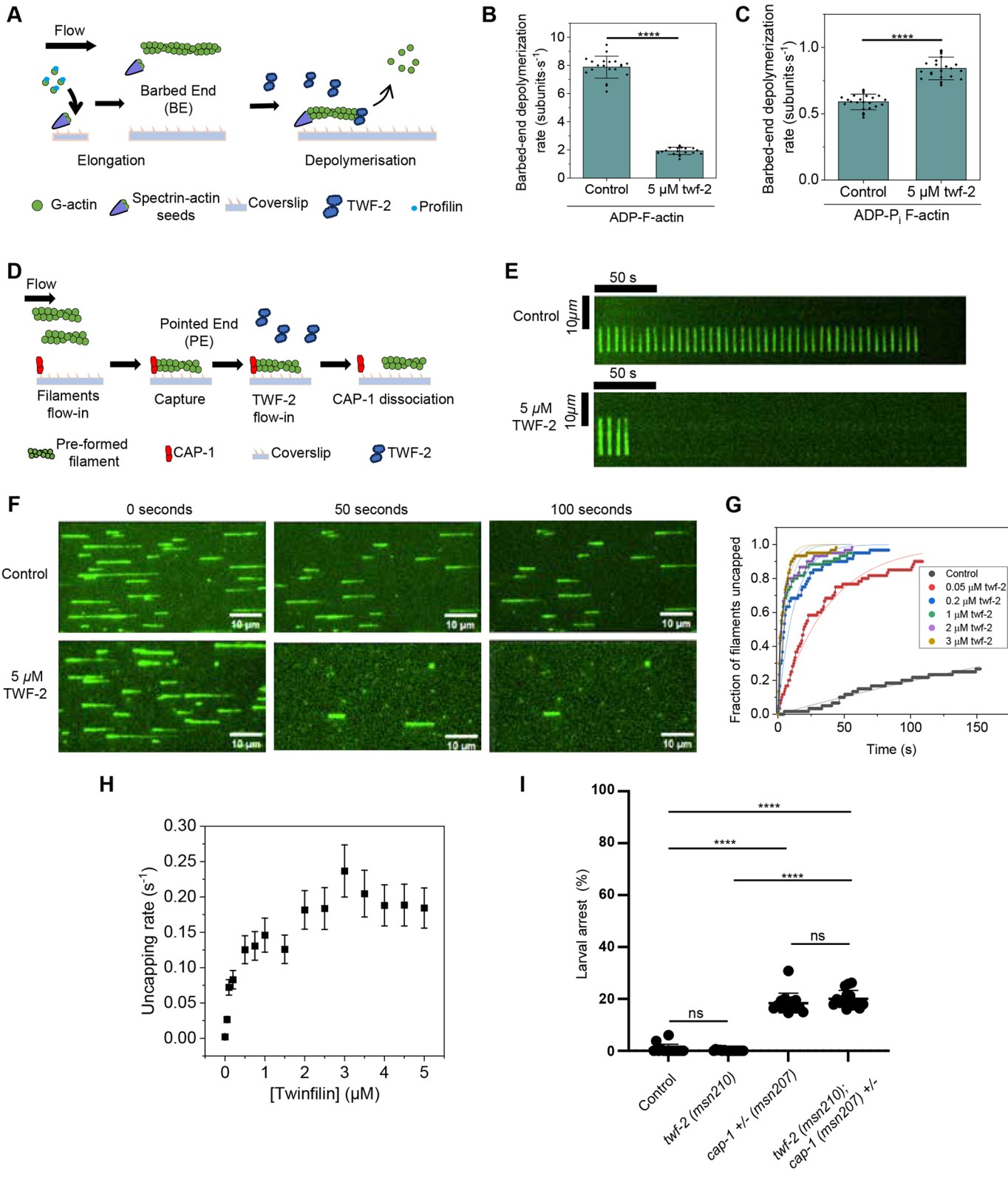

**Fig. 3.** See next page for legend.

yielded a 54% reduction in embryonic lethality (30.4±13% in control versus 13.9±11% in *twf-2 RNAi*; Fig. 5B). A genetic cross producing *twf-2;spv-1* double mutants confirmed this genetic interaction, with embryonic lethality reduced by 43% (28.4±11%) compared to *spv-1* single mutants (49.96±14%; Fig. 5C). To confirm the spermathecal

origin of this rescue, we performed *twf-2* RNAi in a spermatheca-specific RNAi strain, in an *spv-1* mutant background, and observed a similar reduction in embryonic lethality phenotype (Fig. 5E).

The hypercontractility of *spv-1* mutant spermathecae manifests itself in abnormal embryonic morphology due to abnormal

**Fig. 3. Effect of TWF-2 on barbed-end depolymerization and uncapping of CP from actin filaments.** (A) Schematic representation of the experimental strategy. Actin filaments with free barbed ends were polymerized by exposing coverslip-anchored spectrin-actin seeds to 1 µM G-actin (15% Alexa-488 labeled) and 4 µM profilin. To generate ADP-actin filaments, the filaments were first aged for 15 min and then exposed to TWF-2 in TIRF buffer. For experiments with ADP-$P_i$ filaments, 50 mM excess phosphate was added to all reactions and filaments were not aged. Barbed-end depolymerization was monitored over time. (B) Rates (±s.d.) of barbed-end depolymerization of ADP-actin filaments in the presence of buffer (control) or 5 µM TWF-2. We analyzed 20 filaments for each condition. Statistical significance was determined by Mann–Whitney test (****$P$<0.0001). (C) Rates (±s.d.) of barbed-end depolymerization of ADP-$P_i$-actin filaments in the presence of buffer (control) or 5 µM TWF-2. We analyzed 25 filaments for each condition. Statistical significance was determined by Mann–Whitney test (****$P$<0.0001). (D) Schematic representation of the experimental strategy. Preformed Alexa-488 labeled ADP-actin filaments were captured by coverslip-anchored biotinylated SNAP-CP. Time-dependent detachment of filaments at a range of TWF-2 concentrations was monitored. (E) Representative kymographs of a CP-anchored filament uncapping in the presence of buffer (top) or 5 µM TWF-2 (bottom). (F) Representative mf-TIRF time-lapse images showing filament departure due to uncapping. Filaments were exposed to either buffer (top) or 5 µM TWF-2 (bottom). Scale bar: 10 µm. (G) Fraction of filaments uncapped as a function of time in the presence of a range of TWF-2 concentrations. Experimental data (symbols) are fitted to a single-exponential function (lines). We analyzed 59-60 filaments for each condition. (H) CP dissociation rate as a function of TWF-2 concentration, determined from data shown in G. (I) Larval arrest quantifications. Dot plots indicate percentage larval arrest for control (wild type), *twf-2(msn210)* mutant, *cap-1(msn207)* mutant and *cap-1(msn207); twf-2(msn210)* double mutant strains (*n*≥14). Statistical significance was determined by Kruskal–Wallis test (ns, not significant; ****$P$<0.0001). Data are mean±s.d. CP, capping protein.

squeezing and pinching of the embryos while they transit through the spermatheca (Tan and Zaidel-Bar, 2015). Morphometric analysis revealed that *twf-2* loss significantly restored embryo shape in *spv-1* mutants, with embryo area and axial ratio distributions returning toward wild-type values (Fig. 5D,F-I).

To directly assess the impact of *twf-2* loss on spermathecal hypercontractility of *spv-1* mutants, we recorded and analyzed ovulation events in control and both single and double mutant strains, using differential interference contrast (DIC) microscopy (Fig. 6A). *spv-1* loss is known to reduce the time an oocyte remains in the spermatheca, defined as the time between closure of the distal spermatheca valve and opening of the sp-ut valve ('valve-to-valve time'). The hypercontractility of the spermatheca in *spv-1* mutants frequently pinches the oocyte/embryo during its entry or exit, leading to embryonic lethality. Surprisingly, the *twf-2;spv-1* double mutant did not exhibit any significant improvement in valve-to-valve time (Fig. 6B).

Since the pinching of embryos can happen during entry or exit from the spermatheca, we quantified the distal valve transit time and sp-ut valve transit times (Fig. 6C,D), defined as the duration between the opening and closing of the respective valve to allow oocyte entry and exit, respectively. *spv-1* mutants displayed a significantly prolonged distal valve transit time (135±49.54 s) compared to control worms (43.64±18.27 s in N2 and 57.5±19.34 s in *twf-2* mutant). Notably, the *twf-2 spv-1* double mutant exhibited a significant reduction in distal valve transit time (76.93±27.13) compared to the *spv-1* single mutant.

The reduction in distal valve transit time suggests decreased resistance to the incoming oocyte, likely due to restored contractility of the spermatheca bag. These findings indicated

that *twf-2* loss mitigates embryonic pinching and lethality in *spv-1* mutants specifically by facilitating faster oocyte entry through the distal valve.

## *twf-2* loss rescues increased F-actin levels in the spermatheca of *spv-1* mutants

Having established that loss of *twf-2* alleviates the hypercontractility of *spv-1* mutants at the tissue and functional levels, we next investigated the underlying molecular basis of this interaction by examining the actomyosin cytoskeleton. To this end, we performed Phalloidin and Phosphomyosin staining on extruded spermathecae from wild-type (N2), *twf-2* mutants, *spv-1* mutants and *twf-2;spv-1* double mutants. Our analysis revealed that *spv-1* loss caused a significant increase in F-actin and phosphorylated myosin levels in the spermatheca, as expected from its role as a negative regulator of RHO-1 (Fig. 7A-C). Remarkably, loss of *twf-2* completely rescued the increased F-actin levels in *spv-1* mutants (Fig. 7B), while phosphorylated myosin levels remained elevated (Fig. 7C). We also quantified endogenous non-muscle myosin II (NMY-1) expression across control, single and double mutant strains, and found no significant changes compared to wild type (Fig. S4). These results demonstrate that TWF-2 specifically regulates F-actin levels in the spermatheca – likely via CP uncapping – without directly influencing myosin expression or activity.

## DISCUSSION

In this study, we provide the first comprehensive characterization of *C. elegans* TWF-2, the sole twinfilin ortholog in this organism. We show that TWF-2 is broadly expressed and acts as a modulator of spermathecal contractility through its CP uncapping activity. TWF-2 exhibits cortical localization, which in the spermatheca depends on α-spectrin (SPC-1) and β-spectrin (UNC-70). Supporting this association, spectrins were identified in our pulldown of twinfilin, and a previous IP-MS study (Jia et al., 2020) detected TWF-2 (annotated as F38E9.5) as a potential SPC-1 interactor. Whether the interaction is direct or mediated through additional factors remains an open question. The dependence of TWF-2 cortical localization on spectrins highlights specialized mechanisms for spatial control of actin dynamics in distinct tissues.

Although TWF-2 and CP functionally interact to regulate spermathecal contractility – as reflected in embryo morphology – their localization does not always coincide, suggesting context-dependent roles. CP is known to function independently of TWF-2, for example in the dynein–dynactin complex, and TWF-2 may also have CP-independent functions yet to be uncovered. These observations highlight the multifunctional nature of both proteins and emphasize that their interplay is tightly regulated in a tissue- and process-specific manner.

Our *in vitro* assays confirmed that *C. elegans* TWF-2 retains the conserved ability to uncap CP and that its barbed-end depolymerization activity is nucleotide-state dependent, resembling that of mouse twinfilin (Shekhar et al., 2021). The concentration-dependent uncapping kinetics observed *in vitro* suggest that tissue-specific expression levels of TWF-2, as we observed, may be crucial for tuning its activity *in vivo*. Strikingly, whereas mouse twinfilin alone enhances uncapping only sixfold (Hakala et al., 2021; Reddy et al., 2025), *C. elegans* TWF-2 accelerates it by over 90-fold, pointing to an evolutionary adaptation for rapid uncapping in this species. These biochemical activities provide a mechanistic basis for our genetic findings: loss of *twf-2* partially rescued the embryonic

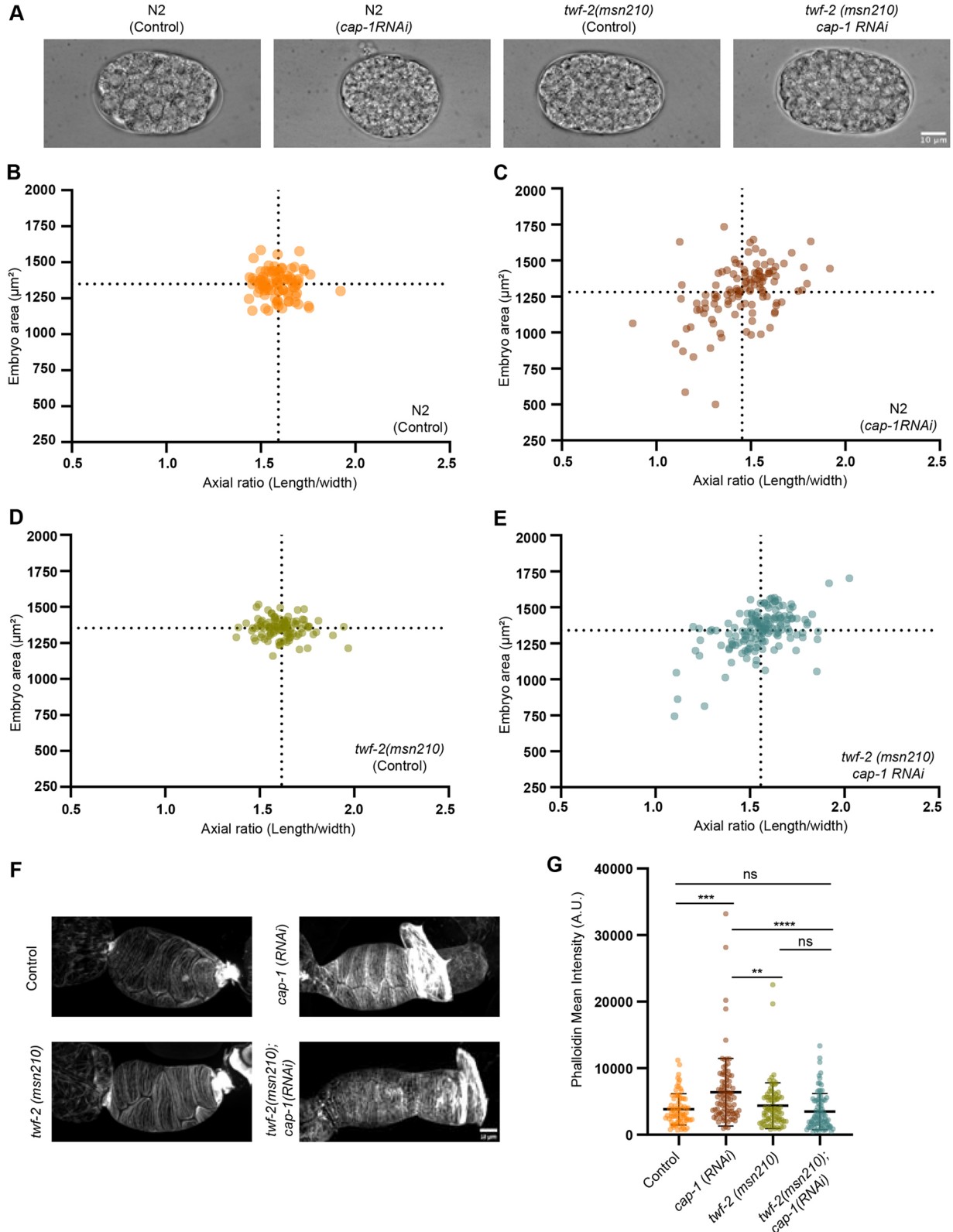

**Fig. 4. Genetic interaction between TWF-2 and CAP-1 impacts embryonic viability and morphology.** (A) Representative DIC images showing embryo morphology (area and axial ratio) in wild-type (N2) control, *cap-1* RNAi in N2, *twf-2(msn210)* control*,* and *cap-1* RNAi in *twf-2(msn210)*. Scale bar: 10 μm. (B-E) Quantitative analysis of embryo morphology. Correlation plots show embryo area versus axial ratio in wild-type (N2) control, *cap-1* RNAi in N2, *twf-2(msn210)* control, and *cap-1* RNAi in *twf-2(msn210)*. Dotted lines indicate the mean values. *n*≥87 embryos per genotype. (F) Representative maximum *z*-projection confocal images of extruded spermathecae stained for F-actin (Phalloidin) in wild-type (N2) control, *cap-1* RNAi in N2, *twf-2(msn210)* control, and *cap-1* RNAi in *twf-2(msn210)*. Scale bar: 10 μm. (G) Quantification of mean fluorescence intensity of F-actin in extruded spermathecae in wild-type (N2) control, *cap-1* RNAi in N2, *twf-2(msn210)* control, and *cap-1* RNAi in *twf-2(msn210)*. Statistical significance was determined by Kruskal–Wallis ANOVA test (ns, not significant; \*\**P*<0.01, \*\*\**P*<0.001, \*\*\*\**P*<0.0001). *n*≥76 spermathecae cells per genotype from three biological replicates were quantified. Data are mean±s.d.

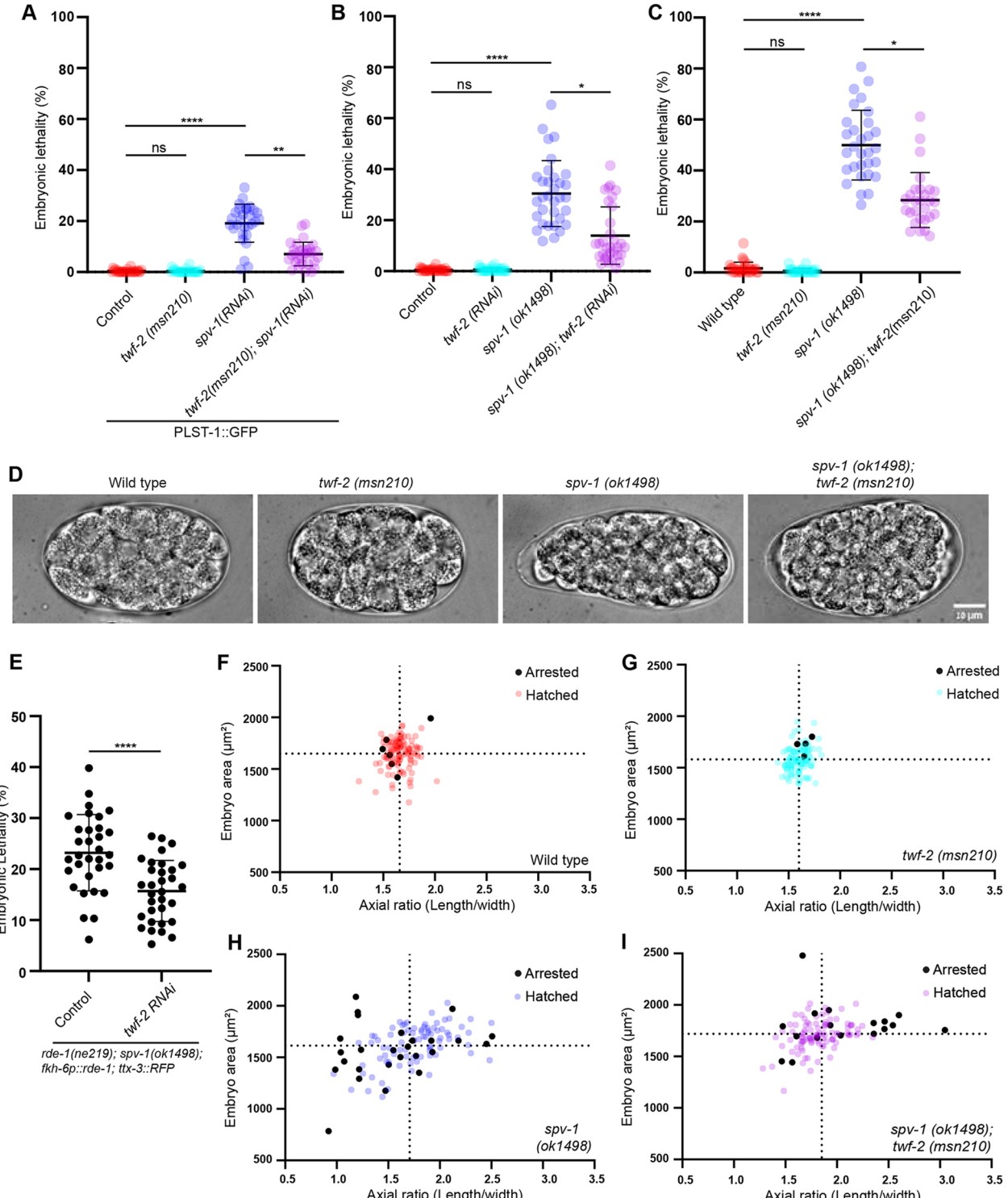

Fig. 5. Genetic interaction between TWF-2 and SPV-1 impacts embryonic viability and morphology. (A) Embryonic lethality in control and *twf-2(msn210)* mutant subjected to systemic *spv-1 RNAi*. Statistical significance was determined by Kruskal–Wallis ANOVA test (ns, not significant; \*\**P*<0.01, \*\*\*\**P*<0.0001). *N*=3 biological replicates with *n*≥2700 embryos scored per condition. (B) Embryonic lethality in control and *spv-1(ok1498)* mutant upon systemic *twf-2 RNAi*. Statistical analysis as in A (ns, not significant; \**P*<0.05, \*\*\*\**P*<0.0001). *N*=3 biological replicates with *n*≥2500 embryos scored per condition. (C) Embryonic lethality in control, *twf-2(msn210)* mutant, *spv-1(ok1498)* mutant *and twf-2(msn210);spv-1(ok1498)* double mutant strains. Statistical analysis as in A (ns, not significant; \**P*<0.05, \*\*\*\**P*<0.0001). *N*=3 biological replicates with *n*≥2200 embryos scored per condition. Data are mean±s.d. (D) Representative DIC images showing embryo morphology (area and axial ratio) in wild-type, *twf-2(msn210)*, *spv-1(ok1498)* and *twf-2(msn210); spv-1(ok1498)* double mutant strains. Scale bar: 10 µm. (E) Tissue-specific rescue using spermatheca-restricted RNAi. Dot plot shows embryonic lethality in *spv-1(ok1498)* mutant, upon control and *twf-2 RNAi*. Statistical significance was determined by Mann–Whitney test (\*\*\*\**P*<0.0001). *n*≥2500 embryos were scored for each condition. Data are mean±s.d. (F-H) Quantitative analysis of embryo morphology. Correlation plots show embryo area versus axial ratio in wild-type (F), *twf-2(msn210)* (G), *spv-1(ok1498)* (H) and *twf-2(msn210);spv-1(ok1498)* double mutant (I) strains. Black points denote arrested embryos; colored points denote hatched embryos. Dotted lines indicate the mean values. *n*≥109 embryos per genotype.

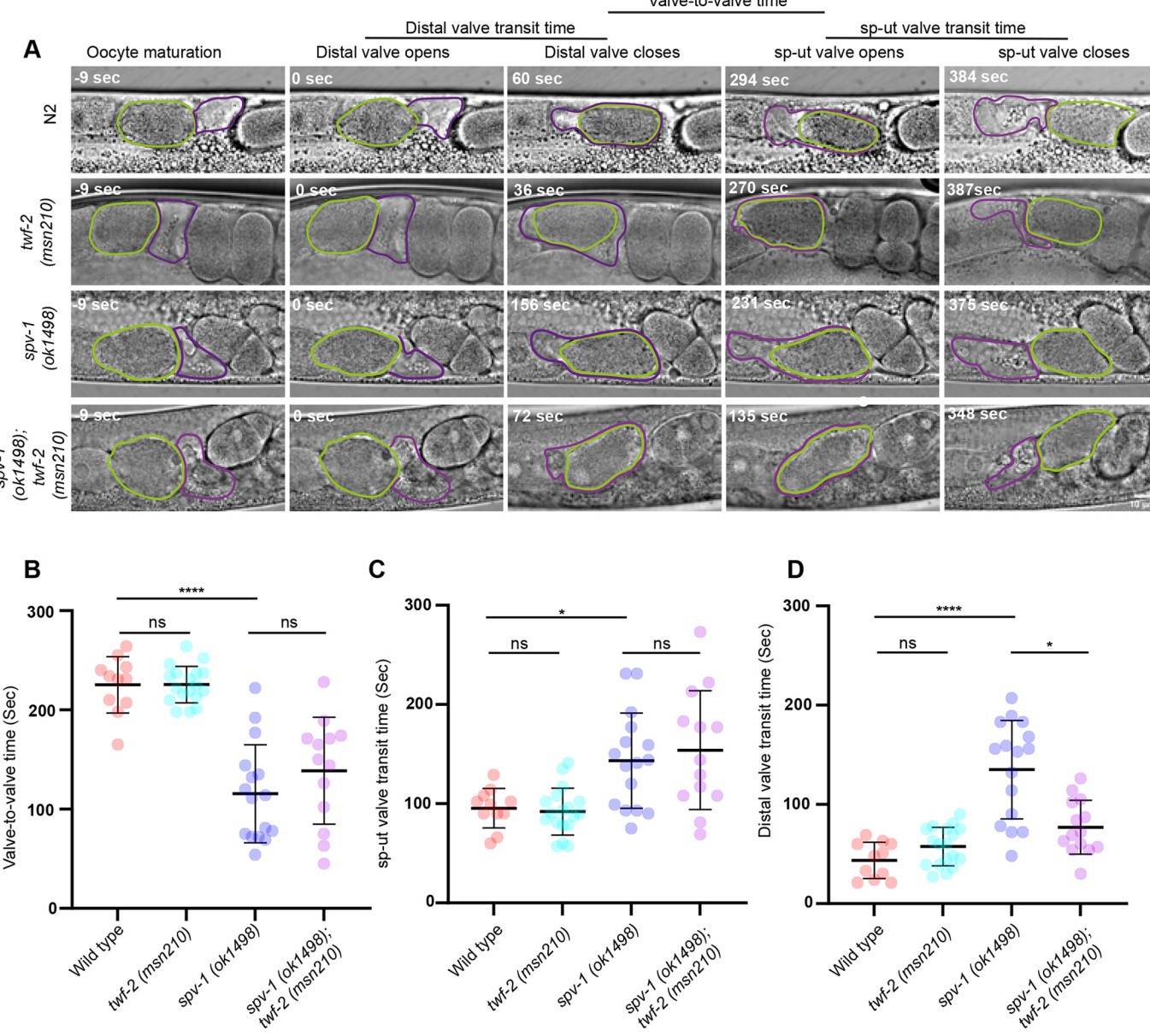

**Fig. 6. Genetic interaction between TWF-2 and SPV-1 modulates spermathecal contractility.** (A) Representative ovulation and embryo transit events in wild-type (N2), *twf-2(msn210)*, *spv-1(ok1498)* and *twf-2(msn210);spv-1(ok1498)* double mutant strains. Opening of the distal valve is referred to as time 0. Valve-to-valve time is the interval between distal valve closure and opening of the spermatheca-uterine valve. Oocyte is marked by green line and spermatheca is marked by purple line. Scale bar: 10 µm. (B-D) Quantification of valve-to-valve time (B), sp-ut valve transit time (C) and distal valve transit time (D) in wild-type, *twf-2(msn210)*, *spv-1(ok1498)* and *twf-2(msn210);spv-1(ok1498)* double mutant strains. Data are represented as individual embryo transit events with mean±s.d. Statistical significance was determined by Kruskal–Wallis ANOVA test (ns, not significant; *$P<0.05$, ****$P<0.0001$). $n \geq 11$ for each worm strain analyzed.

lethality and completely rescued the elevated F-actin levels caused by *cap-1* RNAi, thereby counteracting spermathecal hypercontractility. Conversely, no rescue was observed in *cap-1; twf-2* double mutants, confirming that a minimal level of CP is required for TWF-2 loss to have an effect and supporting a direct functional antagonism between TWF-2 and CP *in vivo*.

The physiological relevance of TWF-2-mediated actin regulation in the spermatheca is further supported by its genetic interaction with the RhoGAP SPV-1, which is expressed exclusively in this tissue. We found that *twf-2* loss alleviates the embryonic lethality and morphology defects of *spv-1* mutants, and fully rescues their elevated F-actin levels. Importantly, while *twf-2* loss reduced

phosphorylated myosin levels, this did not result in overt changes in spermathecal contractility, as judged from ovulation movies, suggesting functional compensation under wild-type conditions. Furthermore, loss of *twf-2* did not rescue the increase in non-muscle myosin II expression and phosphorylation levels found in the *spv-1* mutant, reflecting a role for TWF-2 strictly related to actin and not myosin. This is in contrast to findings in mouse cochlear mechanosensory stereocilia, where twinfilin affects both actin and myosin regulation (Rzadzinska et al., 2009). Such uncoupling of actin and myosin control in the spermatheca suggests that TWF-2 functions as a specialized modulator of actin assembly, likely tuned to respond to mechanical cues associated with ovulation.

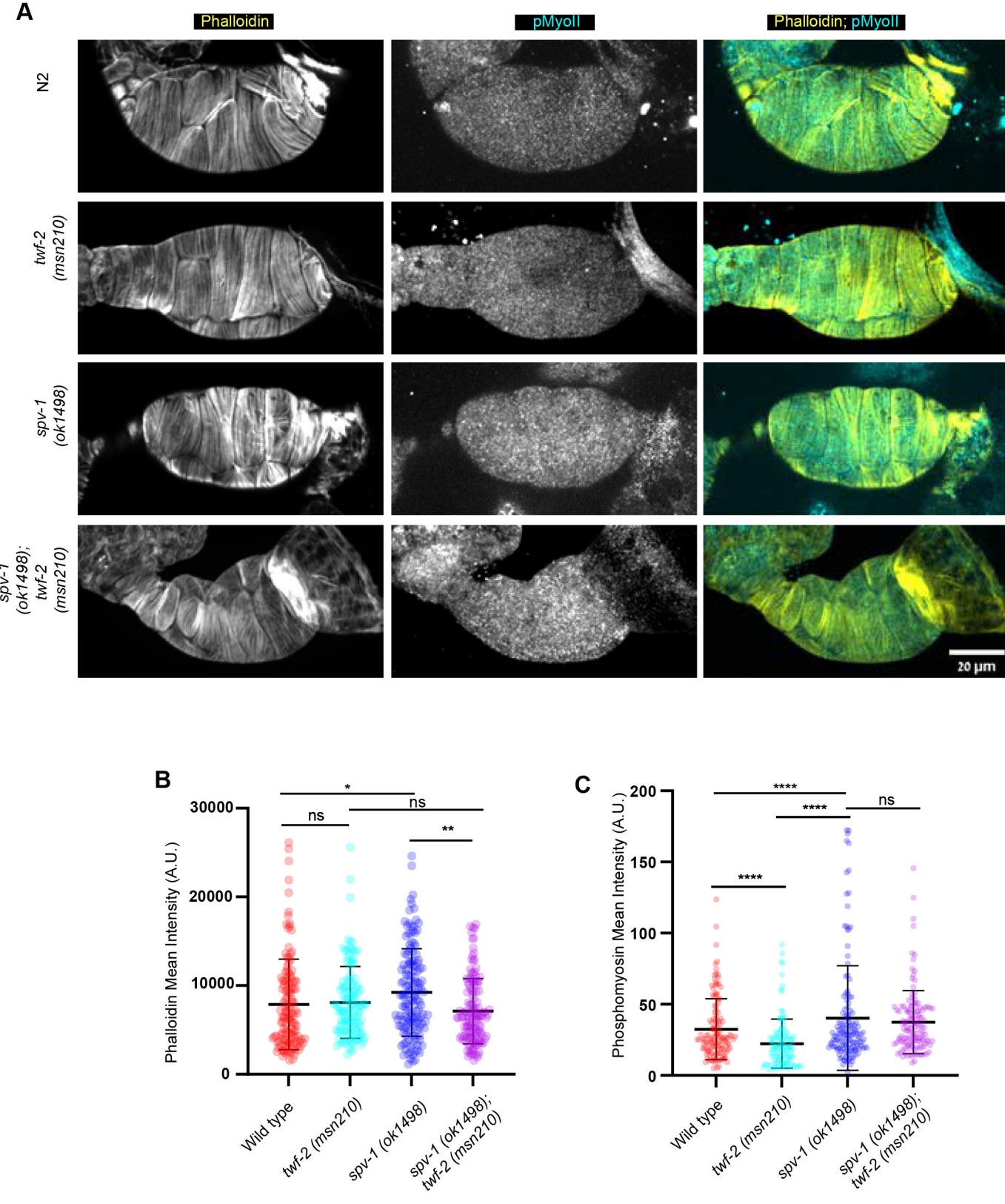

**Fig. 7. TWF-2 regulates F-actin levels in the spermatheca.** (A) Representative confocal images of extruded spermathecae stained for F-actin (Phalloidin, yellow) and phosphorylated myosin (cyan) in wild-type, *twf-2(msn210)*, *spv-1(ok1498)* and *twf-2(msn210);spv-1* double mutant. Scale bar: 20 µm. (B) Quantification of mean fluorescence intensity of F-actin in extruded spermatheca of wild-type, *twf-2(msn210)*, *spv-1(ok1498)* and *twf-2(msn210);spv-1* double mutant. Statistical significance was determined by Kruskal–Wallis ANOVA test (ns, not significant; *$P<0.05$, **$P<0.01$). $n\geq123$ spermathecae cells per genotype from three biological replicates were quantified. (C) Quantification of mean fluorescence intensity of phosphomyosin in extruded spermatheca of wild-type, *twf-2(msn210)*, *spv-1(ok1498)* and *twf-2(msn210);spv-1* double mutant. Statistical significance was determined as in B (ns, not significant; ****$P<0.0001$). $n\geq123$ spermathecae cells per genotype from three biological replicates were quantified. Data are mean±s.d.

Together, our findings bridge molecular biochemistry with tissue-level physiology, demonstrating how the activities of TWF-2 – CP uncapping and nucleotide-state-sensitive depolymerization – translate into functional control of contractility. The spectrin–TWF-2–CP axis we describe likely represents a broader mechanism of actin regulation at the cell cortex, with implications for other contractile systems. The *C. elegans* spermatheca, with its genetic tractability and quantifiable mechanical outputs, provides an ideal model to investigate how actin-binding proteins coordinate contractility. More broadly, our work establishes twinfilin as a potent physiological uncapper that links spectrin-based cortical organization to CP-mediated actin regulation, uncovering a conserved but previously unappreciated mechanism for fine-tuning actomyosin contractility in living tissues.

## MATERIALS AND METHODS

### Worm strain maintenance
All worm strains were maintained at 15°C, as per standard worm protocols (Brenner, 1974). All the experiments were performed at 20°C. All strains used in this study are listed in Table S2.

### CRISPR-Cas9-mediated generation of reporter and knockout strains
The TWF-2 endogenous reporter strain and *twf-2* null mutant were generated using CRISPR-Cas9-mediated genome editing (Paix et al., 2017). To create an endogenous reporter strain, *twf-2* was tagged at the N-terminus with GFPnovo2 (Hendi and Mizumoto, 2018). A single guide RNA (sgRNA) (aggagaagaagctactactA) was designed to insert the GFP fluorophore, along with a linker sequence (TCTGGTGGTAGTGGCGGT ACC), downstream of the 5′ untranslated region (UTR) and in frame with the first exon. The repair template, consisting of GFP and the linker sequence, was PCR-amplified from a plasmid and included 35 bp homology arms flanking the 5′ UTR and the first exon. The template was purified using the Macherey-Nagel kit and eluted in water.

The injection mix contained Cas9 protein (0.8 μg/μl; Integrated DNA Technologies), *twf-2* sgRNA (0.0625 nmol/μl), GFP repair template (4000 ng), *dpy-10* sgRNA (0.02 μg/μl), *dpy-10* ssODN (0.05 μg/μl), KCl (25 mM), tracrRNA (0.1 μg/μl) and water in a total volume of 20 μl. The mix was microinjected into the gonads of young adult N2 hermaphrodites.

F1 progeny with the dumpy (*dpy-10*) phenotype were isolated onto individual nematode growth medium (NGM) agar plates, allowed to lay embryos, and then genotyped via single-worm lysis followed by PCR. Genotyping was performed using primers (ccgaacatttggatgggaagg and agtcaacacccacttaacccc). The insertion of GFP was confirmed through DNA sequencing.

For generation of a null mutant, two sgRNAs were designed to delete a 2840 bp region of the *twf-2* gene. The first sgRNA (CTTCGAAATGCAC TCAACTT) targeted exon 2 while the second sgRNA (aaattaaatattgcagAT GG) targeted exon 8. A single-stranded oligodeoxynucleotide (ssODN) repair template (tttgcagCTAACGCGGCACTTCGAAATGCACTCAAT GGAGGTAGATGCTCGCGACGATCTTTCGGAGAAA) was synthesized with 35 bp homology arms flanking the deletion site [lowercase letters are for noncoding sequences (UTRs and introns) and uppercase letters represent coding region (exons)].

The injection mix contained Cas9 protein (0.8 μg/μl), *twf-2* sgRNAs (0.0625 nmol/μl), *twf-2* ssODN (0.225 μg/μl), *dpy-10* sgRNA (0.02 μg/μl), *dpy-10* ssODN (0.05 μg/μl), KCl (25 mM), tracrRNA(0.1 μg/μl) and water in a total volume of 20 μl. This mix was microinjected into the gonads of young adult N2 hermaphrodites.

F1 progeny exhibiting the dumpy (*dpy-10*) phenotype were isolated onto individual NGM plates and allowed to lay embryos. These F1 animals were lysed and genotyped by PCR to screen for the deletion. Specific primers (gtcccagacacacttctcttcc and tcgggtcgacgtttcagtatg) were used to identify mutants, while other primers (CTCGTTGCCATCATCTGGAAG and tcgggtcgacgtttcagtatg) were used to confirm the wild-type genotype. The targeted deletion was further validated by DNA sequencing of the progeny.

### RNAi knockdown
Systemic and spermatheca-specific RNAi-mediated gene knockdown were performed by feeding *E. coli* HT115 bacteria expressing double stranded RNA (dsRNA) targeting the gene of interest. RNAi clones *spv-1* and *twf-2* were used with their corresponding empty vector clones (T444T and L4440) as negative controls. RNAi plates were prepared with NGM containing 1 mM IPTG and 100 μg/ml ampicillin.

Primary cultures of HT115 clones were grown in 5 ml Luria-Bertani (LB) broth containing 100 μg/ml ampicillin overnight at 37°C. Secondary cultures were inoculated by adding 200 μl of the primary culture into 20 ml LB containing 100 μg/ml ampicillin and incubated for 7-8 h at 37°C until the culture reached an optical density (OD$_{600}$) of 1. The bacterial culture was centrifuged, and the pellet was resuspended in 2 ml M9 buffer. RNAi plates were seeded with 150 μl of the bacterial suspension and allowed to dry overnight at room temperature.

Adult hermaphrodites of the required genotype were bleached, and embryos were dropped onto the RNAi-seeded plates, after adding 100 μl 100 mM IPTG on the seeded bacteria to induce dsRNA expression. All RNAi experiments in this study were performed at 20°C. Phenotypic analysis was performed on either the same generation or their progeny.

### Embryonic lethality, larval arrest and embryonic shape assays
Embryonic lethality assays were performed using 1-day-old adult hermaphrodites. Ten worms were placed on each RNAi-seeded plate or OP50-seeded plate for 3-4 h, yielding an average of ~100 embryos per plate. After the egg-laying period, the adults were removed and the total number of embryos were counted. The plates were incubated at 20°C for 24 h. Following incubation, the embryos were scored for viability by counting the number of hatched and unhatched embryos. Larval arrest was scored after an incubation of 48 h.

Embryo shape and area were analyzed by dissecting 1-day-old adult hermaphrodites. The embryos were mounted on 3% agarose pads prepared on glass slides, sealed with wax after the addition of M9 buffer. Each embryo was numbered and imaged for shape analysis.

Following imaging, the slides were incubated overnight at 20°C. Embryo viability was assessed 16-20 h post-incubation by scoring hatching.

### Phalloidin staining
Worms were collected in M9 buffer and their gonads were extruded by making an incision within 5 min after addition of 100 mM Levamisole. Fixation was performed by adding 4% formaldehyde in PBS for 15 min. The formaldehyde was removed by centrifugation at 1500 ***g*** for 30 s and ice-cold acetone (−20°C) was added for 5 min.

The samples were washed thrice with PBST (1× PBS containing 0.5% Triton X-100). Phalloidin conjugated to TRITC was added to a final dilution of 1:250 in PBST and incubated at room temperature for 2 h. Post incubation, the samples were washed thrice with PBST, and DAPI was added to a dilution of 1:1000 in PBST, followed by a 30-min incubation at room temperature.

The samples were washed again with PBST, and Vectashield mounting media was added to the samples, which were either stored at 4°C for up to a week or immediately mounted for imaging.

### Immunofluorescence
Worm gonads were extruded in M9 buffer containing 100 mM Levamisole within 5 min. The dissected gonads were fixed with 4% formaldehyde in PBS for 15 min, followed by three washes in PBS. Permeabilization was performed using 0.25% Tween 20 in PBS for 10 min, followed by three washes with PBS.

The samples were incubated in a blocking solution comprising 1% bovine serum albumin (BSA), 0.1% Tween 20 and 30 mM glycine in PBS for 1 h at room temperature. Primary antibody incubation was carried out overnight at 4°C using anti-phospho-MLC (Ser19) (Cell Signaling Technology, #3671) at a 1:400 dilution in the blocking solution.

Gonads were washed thrice with PBS and incubated with blocking buffer containing 1:500 anti-rabbit secondary antibody conjugated with Alexa 488 (Invitrogen, A21244), 1:250 Phalloidin-TRITC (Sigma-Aldrich, P1951) and 1:1000 DAPI (Sigma-Aldrich) at room temperature for 90 min.

Following three washes with PBS, Vectashield mounting media (Vector Laboratories, H-1000) was added. The gonads were stored at 4°C for up to a week or directly mounted for imaging.

## Image acquisition

Imaging was performed using a Nikon Ti-2 Eclipse inverted microscope equipped with a Yokogawa spinning-disk confocal system (CSU-W1) and Plan-Apochromat oil-immersion objectives (60×, 1.4 NA or 100×, 1.45 NA). Samples were illuminated with 405 nm, 488 nm and 561 nm lasers (Gataca Systems) and captured using a Prime 95B sCMOS camera (Photometrics). MetaMorph software (Molecular Devices) was used for acquisition control. All imaging was performed at 20°C.

For fluorescence and DIC microscopy, worms were mounted on 3% agarose pads and immobilized using Levamisole. Embryos were mounted in M9 buffer and imaged using 100×1.45NA oil-immersion objective.

For ovulation movies, worms were mounted in 1.5 µl M9 buffer, on 9% agarose pads and covered using a cover slip. DIC images were acquired at 60× magnification, at a time interval of 3 s.

## Image analysis and statistics

All image analyses were performed using ImageJ (National Institutes of Health) (Schindelin et al., 2012). Regions of interest (ROIs) were manually selected for subsequent quantification. For co-localization analysis (Fig. 1), a 5-pixel-wide line was drawn along the ROI for both channels. The mean intensity was calculated and background intensity was subtracted. The local maxima of one fluorophore peak was identified using Find Peaks FIJI plugin and aligned across different images manually using Microsoft Excel. The values were plotted and Pearson correlation coefficients were determined using GraphPad Prism 10.

Embryo shape and area were quantified by manually tracing the eggshell using the segmented line tool in ImageJ. Embryo length and width were measured by drawing straight-line segments along the longest and shortest axes, respectively.

Phalloidin quantifications were performed by drawing a 20-pixel-wide line at a specific z-stack to measure the mean intensity of actin fibers in a single cell. A maximum of two cells were analyzed per individual spermatheca. Phosphomyosin mean intensity was measured in the same ROI, using the same 20-pixel-wide line as for the Phalloidin measurements. For both stainings, background intensity was subtracted for each image to ensure accurate quantification.

Quantifications of ovulation movies were carried out manually. Dwell time was defined as the duration an oocyte remained in the spermatheca, measured from the closure of the distal valve to opening of the sp-ut valve. Transit times were calculated separately for the distal valve and sp-ut valve, as the time between their opening and closing.

Statistical analyses were performed using GraphPad Prism software. Briefly, quantitative datasets were subjected to normality tests to define datasets as parametric or non-parametric. For comparison between two datasets, unpaired t-test was performed and for comparison among three or more groups, one way ANOVA was performed.

## Immunoprecipitation and mass spectrometry

Synchronized GFP::TWF-2 knock-in strains and N2 L1 larvae were cultured on ~10, 60-mm NGM plates at 20°C. After 50-52 h, mixed populations were collected and washed three times with M9 buffer (42.33 mM $Na_2HPO_4$, 22.06 mM $KH_2PO_4$, 85.56 mM NaCl, 1 mM $MgSO_4$). Worms were fixed with 0.5% paraformaldehyde in M9 buffer for 20 min at 20°C. The fixative was removed, and worms were incubated with 50 mM Tris (pH 8.0) in M9 buffer for 5 min at 20°C. After two washes with M9 buffer, the worm pellet was stored at −80°C.

Frozen samples were thawed on ice and washed with cold 1× PBS. Worms were resuspended in ~500 µl of lysis buffer [50 mM Tris-HCl (pH 7.4), 150 mM NaCl, 1 mM EDTA, 1% Triton X-100] containing a protease inhibitor tablet. Sonication was performed on ice until worm bodies were no longer visible in the buffer. The lysate was centrifuged at 15,000 rpm (12,000 $g$) for 20 min at 4°C, and the supernatant was transferred to new tubes. A 30 µl aliquot was set aside for Bradford analysis and western blotting, while the remaining lysate was stored at −80°C.

Immunoprecipitation was carried out using G-Sepharose beads. For pre-clearance, 15 µl of protein G beads were added to the lysate and incubated at 4°C for 2 h. After centrifugation at 12,000 $g$ for 15 min at 4°C, the supernatant was transferred to fresh tubes and incubated overnight with 8 µg of anti-GFP mouse antibody (Roche, 11814460001, 0.4mg/ml stock) at 4°C. Following antibody binding, 15 µl of beads were added, and the samples were incubated at 4°C for 3 h. Beads were collected by centrifugation at 12,000 $g$ for 30 s at 4°C and washed three times with 1× wash buffer [0.05 M Tris-HCl (pH 7.4), 0.15 M NaCl]. A final wash was performed with PBS containing 1% Triton X-100.

The samples were boiled in 1× Laemmli sample buffer at 95°C for 10 min. After centrifugation at 15,000 rpm (12,000 $g$), the supernatant was transferred to clean tubes. We used 10% of the sample for SDS-PAGE western blotting as a quality check, and the remaining sample was subjected to western blotting followed by trypsin digestion and analysis by liquid chromatography-tandem mass spectrometry on Q-Exactive HF (Thermo Fisher Scientific). Mass spectrometry was performed at the Smoler Protein Research Center at the Technion Institute of Technology.

## Protein purification for in-vitro experiments

Proteins were purified as per the established protocol (Shekhar et al., 2021). Briefly, rabbit skeletal muscle actin was purified from acetone powder (PelFreez). Lyophilized powder was sheared, resuspended in G-buffer [5 mM Tris-HCl (pH 7.5), 0.5 mM DTT, 0.2 mM ATP, 0.1 mM $CaCl_2$], cleared by centrifugation (50,000 $g$, 20 min) and filtered. Actin was polymerized overnight at 4°C by adding 2 mM $MgCl_2$ and 50 mM NaCl, and then 0.6 M NaCl was added. F-actin was centrifuged (280,000 $g$, 150 min), homogenized, dialyzed against G-buffer (48 h), gel-filtered (Sephacryl S-200) and stored at 4°C.

For labeling, G-actin was polymerized in modified F-buffer [20 mM PIPES (pH 6.9), 100 mM KCl, 0.2 mM $CaCl_2$, 0.2 mM ATP], incubated with Alexa-488 NHS ester (5:1 molar excess, 2 h, room temperature), pelleted (450,000 $g$, 40 min), depolymerized in G-buffer and repolymerized (100 mM KCl, 1 mM $MgCl_2$). After a final ultracentrifugation (450,000 $g$, 40 min), labeled actin was dialyzed, cleared (450,000 $g$, 40 min) and assessed for concentration and labeling efficiency.

C. elegans TWF-2 (6×His-tagged) was expressed in E. coli BL21 (pRare) induced with 1 mM IPTG (18°C, overnight). Cells were lysed in 50 mM $NaPO_4$ (pH 8), 300 mM NaCl, 20 mM imidazole, 1 mM DTT and protease inhibitors, sonicated and cleared (120,000 $g$, 45 min). The lysate was incubated with Ni-NTA beads (2 h, 4°C), washed and eluted with 250 mM imidazole. Purified protein was further resolved by size-exclusion chromatography [Superdex 75 Increase, 20 mM HEPES (pH 7.5), 50 mM KCl, 1 mM DTT], concentrated, flash-frozen and stored at −80°C.

Human profilin 1 was expressed in E. coli BL21 (pRare), induced with 1 mM IPTG. Cells were lysed in 50 mM Tris-HCl (pH 8), 1 mM DTT and protease inhibitors, sonicated and centrifuged (120,000 $g$, 45 min). The supernatant was applied to poly-L-proline beads, washed [10 mM Tris (pH 8), 150 mM NaCl, 1 mM DTT] and eluted with 8 M urea. The protein was dialyzed [2 mM Tris (pH 8), 0.2 mM EGTA, 1 mM DTT], ultracentrifuged (450,000 $g$, 45 min), concentrated, flash-frozen and stored at −80°C.

C. elegans SNAP-tagged CP was expressed in E. coli BL21 DE3 (1 mM IPTG, 18°C, overnight). Cells were lysed in 20 mM $NaPO_4$ (pH 7.8), 300 mM NaCl, 15 mM imidazole, 1 mM DTT and protease inhibitors, sonicated and cleared (150,000 $g$, 30 min). The lysate was purified via HisTrap chromatography, eluted with 250 mM imidazole, and labeled with benzylguanine-biotin. Free biotin was removed by size-exclusion chromatography [Superose 6, 20 mM HEPES (pH 7.5), 150 mM KCl, 0.5 mM DTT]. Purified protein was aliquoted, flash-frozen and stored at −80°C.

## Microfluidics-assisted TIRF microscopy

mf-TIRF microscopy was performed as per the established protocol (Shekhar et al., 2021). Briefly, actin filaments were assembled in mf-TIRF flow cells (Shekhar, 2017), Coverslips were cleaned by sonication in Micro90 detergent and 1 M KOH, 1 M HCl and ethanol, then coated with mPEG-silane (2 mg/ml) and biotin-PEG-silane (2 µg/ml) in acidic 80% ethanol (pH 2.0) and incubated overnight at 70°C. A PDMS flow chamber

(40 μm high, 3 inlets, 1 outlet) was clamped onto the coated coverslip and connected to a microfluidic flow-control system (Fluigent) and rinsed with TIRF buffer. Chambers were incubated with 1% BSA and 10 μg/ml streptavidin in HEPES/KCl buffer.

For uncapping experiments, biotinylated CP was anchored to the coverslip surface, and preformed Alexa-488-labeled actin filaments (15% labeled) were introduced. Filament survival was monitored in TIRF buffer [10 mM imidazole (pH 7.4), 50 mM KCl, 1 mM MgCl$_2$, 1 mM EGTA, 0.2 mM ATP, 10 mM DTT] with and without addition of twinfilin. The cumulative distribution functions were fit to an exponential decay function to determine the barbed-end dissociation rate of CP.

For depolymerization assays, filaments were grown from spectrin-actin seeds using 1 μM G-actin and 4 μM profilin in TIRF buffer, aged to ADP-actin (15 min in 0.1 μM G-actin in TIRF buffer) and then exposed to TIRF buffer with or without twinfilin. Depolymerization rates were measured from kymographs (Fiji). For ADP-P$_i$-actin experiments, filaments were maintained in phosphate-supplemented TIRF buffer (34.8 mM K$_2$HPO$_4$, 15.2 mM KH$_2$PO$_4$).

Images were acquired using a Nikon Ti2000 microscope (60×1.49 NA TIRF objective, Perfect Focus System) with 488/561/640 nm lasers and an EMCCD camera (Andor Ixon 888). Images were acquired with Nikon Elements, drift-corrected (Fiji) and analyzed via kymographs. Rates were averaged from at least three replicates.

### Acknowledgements
Some strains were provided by the Caenorhabditis Genetics Center (CGC), which is funded by National Institutes of Health Office of Research Infrastructure Programs (P40 OD010440).

### Competing interests
The authors declare no competing or financial interests.

### Author contributions
Conceptualization: A.S., R.Z.-B.; Formal analysis: A.S., S.K., E.T., J.M.-L., I.M.Z., L.R., S.S., R.Z.-B.; Funding acquisition: R.Z.-B.; Investigation: A.S., S.K., E.T., J.M.-L., I.M.Z., A.N., L.R.; Methodology: A.S.; Resources: A.N., R.Z.-B.; Supervision: S.S., R.Z.-B.; Validation: A.S.; Visualization: A.S., S.K., E.T.; Writing – original draft: A.S.; Writing – review & editing: S.S., R.Z.-B.

### Funding
This work was supported by the Israel Science Foundation (767/20 to R.Z.-B.) and the National Institute of General Medical Sciences (R35GM143050 to S.S.). Open Access funding provided by Tel Aviv University. Deposited in PMC for immediate release.

### Data and resource availability
All relevant data and details of resources can be found within the article and its supplementary information.

### Peer review history
The peer review history is available online at https://journals.biologists.com/dev/lookup/doi/10.1242/dev.205265.reviewer-comments.pdf

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
