## [Peer Review File · Development (Cambridge, England)]

Twinfilin modulates tissue contractility through uncapping of capping protein in *C. elegans*

Anupreet Saini, Shir Kreizman, Ekram Towsif, Jonathan Martinez-Lopez, Iska Maimon Zielonka, Anat Nitzan, Lee Rudnik, Shashank Shekhar and Ronen Zaidel-Bar
DOI: 10.1242/dev.205265

Editor: Swathi Arur

Review timeline

Submission to Review Commons:	17 June 2025
Submission to Development:	23 September 2025
Editorial decision:	22 October 2025

Reviewer 1

Evidence, reproducibility and clarity

Summary

This manuscript reports characterization of twinfilin in *C. elegans*. Twinfilin is a conserved actin-binding protein and has been characterized in several organisms as an actin-monomer binding protein and a barbed-end depolymerizing protein. TWF-2 is the sole twinfilin in *C. elegans*, and endogenous tagging with GFP showed its expression in multiple tissues. In this work, the authors primarily focused on spermatheca where TWF-2 is expressed and localized to the basolateral cortex of the spermathecal cells. This cortical localization depended on spectrins, SPC-1 and UNC-70. However, *twf-2* knockout worms showed no obvious phenotypes. The *twf-2* knockout partially suppressed embryonic lethality of CAP-1 depletion or SPV-1 knockout, which are interpreted as effects on the contractility of the spermatheca. The *twf-2* knockout reduced F-actin levels in SPV-1 knockout, which may be how the phenotype might be suppressed. These observations led the authors to conclude that TWF-2 is a regulator of cell contractility in the spermatheca. However, this conclusion is not strongly supported by the experimental evidence. The effects on spermathecal contractility is concluded only by embryonic lethality, which is a very indirect parameter. In addition, the *twf-2* knockout causes no detectable phenotypes and only modestly suppresses embryonic lethality in CAP-1 depletion or SPV-1 knockout. Therefore, it is unclear from the current data about how TWF-2 is involved in cell contractility.

Major points

1. Although the conclusion about the TWF-2 function is on the effects on spermathecal contractility, the only presented experimental evidence is the effects on embryonic lethality and deformation, which are considered indirect effects. However, the spermatheca exhibits complex contractility when embryos are transported, and it is not clear how *twf-2* knockout affects this process (contraction or relaxation?). The expression pattern of TWF-2 in Fig. 1 also suggests that TWF-2 is expressed in the spermatheca-uterine valve, which is also an important tissue to allow entry of the embryos to the uterus and may be affected by *twf-2* knockout.
2. Overall, lack of detectable phenotypes in TWF-2 knockout alone and very modest phenotypic modification in CAP-1 RNAi and SPV-1 knockout suggest that TWF-2 as a regulator of contractility is an overstatement.
3. Although SPV-1 is specifically expressed in the spermatheca, CAP-1 is expressed in embryos. The expression pattern of TWF-2 in Fig. 1 suggests that TWF-2 is not expressed in oocytes and embryos. Is this true? Otherwise, TWF-2 and CAP-1 may be functionally related in embryonic development. Effects of *twf-2* knockout on the F-actin organization in the spermatheca of *cap-1* is

not examined, and how twf-2 and cap-1 function together in this tissue is unclear.

4. The authors conclude that TWF-2 knockout suppresses the hypercontractility phenotype of SPV-1 knockout by reducing F-actin without affecting myosin. However, TWF-2 knockout does not affect F-actin in wild-type and indeed reduces phosphomyosin in wild-type (Fig. 5C), suggesting that TWF-2 may affect myosin. Then, what is the interpretation for function of TWF-2 in wild-type background? Also, TWF-2 induces actin depolymerization and removes capping protein from F-actin in vitro (Fig. 3). Therefore, TWF-2 knockout is expected to increase F-actin and stabilize the barbed ends, which is opposite to the in vivo results in the SPV-1 knockout (Fig. 5B). These data that do not align with the conclusion are not discussed well and potentially lead to an alternative conclusions.

5. The observations that the cortical localization of TWF-2 depends on UNC-70 or SPC-1. These are potentially interesting, but what is the relevance to cell contractility in this work?

Minor points

1. Second paragraph of Results. The pharynx is considered a muscle tissue.
2. Table S1B. cdap-2 is duplicated.
3. The ColP proteomics data should be disclosed by depositing to a public database.

Referee cross-comments

I respect the difference in opinions from Reviewer 2. My concerns are not changed. However, I do agree that live imaging of spermatheca contractility would provide more direct evidence for the involvement of TWF-2 in contractility.

Significance

Twinfilin has been studied extensively in vitro, but only in limited model organisms in vivo. The major strengths of this study is the use of a twinfilin-knockout mutant in *C. elegans* and quantitative phenotypic analysis. The in vitro experiments are excellent, but mostly confirms what are already known for the role of twinfilin. The major weaknesses are the lack of detectable phenotypes of the twinfilin knockout worms (when no other mutation/depletion is combined) and very modest phenotypic modifications when capping protein was depleted or a RhoGAP is mutated. Embryonic lethality was interpreted as a consequences of abnormal spermathecal contractility, but the lethality could be caused by other reasons, and the experimental evidence is considered weak. On the other hand, this study makes it clear that twinfilin is not a major regulator of contractility. Therefore, it is not a high-impact study appealing to a broad audience but it is a solid work to report phenotypes and expression patterns of twinfilin in *C. elegans* for limited audience working in the related field.

Reviewer 2

Evidence, reproducibility and clarity

Summary:

The spermatheca must be robust to multiple rounds of stretch and contraction as eggs pass through. The authors show TWF-2 is expressed in contractile tissues, including the spermatheca, where it localizes to the cortex in a spectrin-dependent manner. They show that TWF-2 accelerates depolymeration of actin filaments. They show that loss of twf-2 can partially suppress the embryonic lethality caused by loss of capping protein (CAP-1) and a Rho GAP (SPV-1). This paper shows how twinfillin plays a role in tissue contractility, presumably through modulating F-actin dynamics.

Major comments:

The claims and conclusions are well supported by the data. However, I recommend making it more clear in the abstract that deletion of twf-2 does not cause a phenotype on its own.

Fig. 4 H and I it is not immediately apparent how loss of *twf-2* is suppressing the *spv-1* emb phenotype given these data. I see that there are fewer points in the lower right quadrant. Is this because *spv-1* mutants 'munch' the embryos and the double mutants don't? Perhaps it would help to describe more fully the interpretation of this data in the text.

In Fig. 5, are you certain about the specificity of this pMyoII staining? It looks a bit generally speckly. It would help to include a *mel-11* or myosin light chain knockdown control.

Generally, the data and methods are presented in a way that can be reproduced. The image analysis section should be revised to be more specific about the methods used.

Minor comments:

The description of the statistical analysis (that GraphPad was used) is not adequate. Please generally describe the statistical tests used including trials factor correction methods in the methods section. The figure legends do contain this information, but the methods should have a summary.

I recommend pointing out the reasons you focused on the spermathecal cells for studying subcellular localization of TWF-2 (e.g. the cells are large, exhibit apical basal polarity, exhibit dynamic actin re-arrangements and are contractile).

For completeness, it would be nice to see the spectrin localization in the *twf-2* crispr deletion strain (optional).

In the image analysis section, clarify how the fluorescence intensity was normalized. There are minor editorial issues in the materials and methods, for example, inconsistent spacing between number and unit (should be one space) and some typos including the use of 'u' rather than the 'mu' symbol for micro.

Please indicate what magnification was used for the egg measurements (e.g. 60X).

All legends, check spacing of n {greater than or equal to} x designations. They vary, but most of them need spaces. Also sometimes n is uppercase, sometimes lowercase, and sometimes the N is missing.

In all figures and in text, check spacing between gene and allele designation (e.g. *spv-1(ok1498)*) There should not be a space. Fig. 3 indicate significance on B and C. In the Supplementary Figure 2 legend, *twf-2* should be in italics. 'Null' should not be italicized.

In the Supplementary Figure 3 legend, the word mutant should not be italicized, and there should be a space between wild-type and (N2).

Referee cross-commenting

I agree with some of the concerns of Reviewer #1, however, suggest the authors might be able to address the 'regulator of contractility' claim by modifying the way the results are presented and discussed, rather than undertake a significant series of experiments. SPV-1 is a major regulator of RHO-1 and spermathecal contractility, so the genetic interaction between TWF-2 and SPV-1 does suggest a role in regulation of spermathecal contractility. The reduction in embryonic lethality (loss of the small round eggs, specifically) does suggest TWF-2 regulates contractility in the spermatheca. To more directly show this, the authors could use video imaging to look at spermathecal contractility, as the egg is entering or exiting. Loss of SPV-1 causes a hypercontractile spermathecal neck and sp-ut, these phenotypes could be relieved by loss of TWF-2. I also disagree that the reduction in *cap-1*

RNAi embryonic lethality is 'modest'. It looks pretty significant to me (Fig. 2A), and I'm kind of surprised twf-2 can suppress cap-1 at all.

Significance

- General assessment: The major strength of this study is that it is an in vivo analysis of twinfilin function, showing an important role in contractility that impacts both embryogenesis and reproductive system function in the adult.
- With any study, there is always more that could be done, for example, there are a large number of additional actin-interacting proteins that could be assayed for genetic interactions, or perhaps it makes sense to expand to spectrin interactors. However, this is a nice, clearly presented study showing a role for twinfilin in vivo. While it is surprising that the twinfilin null does not have a phenotype on its own, the genetic interactions make it clear it does have a supporting role.
- Advance: The interaction between spectrin and twinfilin is an important new finding. It is also interesting that the *C. elegans* twinfilin is a particularly potent uncapper of F-actin.
- Audience: This article should be of broad interest in the cytoskeletal (F-actin, spectrin) fields.
- My area of expertise is quite similar to those of the authors of this article.

Reviewer 3

Evidence, reproducibility and clarity

Summary:

Twinfilin is a conserved cytoskeletal regulator, which promotes actin filament uncapping, sustains barbed end depolymerization and sequesters actin monomers. Here, Saini et al. confirm that also *C. elegans* twinfilin (TWF-2) uncaps actin filaments in vitro, similarly to previously shown for mammalian twinfilins. Through elegant genetic studies they reveal that interplay between TWF-2 and Capping protein controls F-actin levels in certain *C. elegans* tissues, including spermatheca, and hence regulates their contractility. Finally, they provide evidence that spectrins are critical for cortical localization of TWF-2 in spermatheca.

The data presented in the manuscript appear of very good technical quality and convincing, and the study extends our knowledge on the physiological functions of twinfilins. However, there are few minor points that should be addressed to strengthen the manuscript.

Minor comments:

1. The authors state in the 'Introduction' that twinfilins accelerate the depolymerization of actin filaments. This is not entirely accurate, because as shown initially by Hakala et al., 2021 and Shekhar et al., 2021 (for some reason the former paper is not cited here, and the latter publication is cited as Shekhar et al., 2020), twinfilin actually inhibits barbed end depolymerization of ADP-actin filaments. However, it can indeed increase depolymerization of ADP-Pi barbed ends, but because twinfilin binds ADP-Pi filament barbed ends with relatively modest affinity and also does not efficiently uncap those, it is unclear if the ADP-Pi barbed end depolymerization is relevant in cells. Perhaps most importantly, both Hakala et al., 2021 and Shekhar et al., 2021 publications demonstrated that twinfilin can sustain actin filament barbed end depolymerization under assembly promoting conditions. Thus, the authors should be more precise here to avoid any confusion.
2. Similarly to above, the authors state in the 'Abstract' and 'Results' that TWF-2 promotes barbed end depolymerization. This again is an overstatement, because based on the data presented in Fig. 3, TWF-2 inhibits barbed end depolymerization of ADP-actin filaments by ~4-fold, and only very modestly enhances barbed end depolymerization of ADP-Pi filaments. Thus, the manuscript text should be revised accordingly.
3. In the 'Introduction' and 'Discussion' the authors state that while mammalian twinfilins alone accelerate filament uncapping by ~6-fold, they synergize with formin mDia1 to accelerate uncapping by over 300-fold. Here, the authors forget to mention that also V-1/myotrophin greatly accelerates the filament uncapping activity of twinfilin (Hakala et al., 2021). This should be clarified in the text.

4. Why is the only twinfilin in *C. elegans* named TWF-2? Could the authors briefly explain/clarify this in the 'Introduction'.
5. Table S1 needs more explanation. For example, why is *cdap-2* listed twice in Table S1B, and why is *cap-1* included in the table (because in the manuscript text the authors state that a homozygous *cap-1* mutant is lethal)? Thus, more text to explain what is actually shown in Table S1 A and S1 B, would be beneficial for the reader.
6. Legend to Fig.1B: 'Single mid-plane' is shown in upper panels (not left) and the 'maximum z-projection' in lower panels (not right).
7. Fig. 3D: Why does the capped filament elongate from its barbed end in the schematic (TWF-2 flow-in)?

Significance

This study advances our understanding on the physiological functions of twinfilin, and will hence be of interest to those studying actin dynamics, as well as to scientists interested in *C. elegans* biology. The biochemical findings on the effects of *C. elegans* twinfilin on actin filament dynamics and uncapping are largely confirmatory, because they show that *C. elegans* twinfilin has similar activities to those reported earlier for mammalian twinfilins. Nevertheless, it was important to demonstrate that these activities are conserved in evolution. Also the interplay between *C. elegans* twinfilin and capping protein *in vivo* is in line with earlier work on yeast and mammalian cells, but the authors have used an elegant set of genetic tools to study this. The most novel finding of the manuscript is the interplay between twinfilin and spectrins, but the underlying mechanism and its possible relevance in other organisms remains to be determined in the future. Thus, the study presents a valuable contribution to the actin dynamics field.

Specific expertise of the reviewer: Actin cytoskeleton biology and biochemistry.

First revision

Author response to reviewers' comments

Manuscript number: RC-2025-03084

Corresponding author(s): Ronen Zaidel-Bar

1. General Statements [optional]
2. Point-by-point description of the revisions

Reviewer #1

Summary

This manuscript reports characterization of twinfilin in *C. elegans*. Twinfilin is a conserved actin-binding protein and has been characterized in several organisms as an actin-monomer binding protein and a barbed-end depolymerizing protein. TWF-2 is the sole twinfilin in *C. elegans*, and endogenous tagging with GFP showed its expression in multiple tissues. In this work, the authors primarily focused on spermatheca where TWF-2 is expressed and localized to the basolateral cortex of the spermathecal cells. This cortical localization depended on spectrins, SPC-1 and UNC-70. However, *twf-2* knockout worms showed no obvious phenotypes. The *twf-2* knockout partially suppressed embryonic lethality of *CAP-1* depletion or *SPV-1* knockout, which are interpreted as effects on the contractility of the spermatheca. The *twf-2* knockout reduced F-actin levels in *SPV-1* knockout, which may be how the phenotype might be suppressed. These observations led the authors to conclude that TWF-2 is a regulator of cell contractility in the spermatheca. However, this conclusion is not strongly supported by the experimental evidence. The effects on spermathecal contractility is concluded only by embryonic lethality, which is a very indirect parameter. In addition, the *twf-2* knockout causes no detectable phenotypes and only modestly

suppresses embryonic lethality in CAP-1 depletion or SPV-1 knockout. Therefore, it is unclear from the current data about how TWF-2 is involved in cell contractility.

Major points

1. Although the conclusion about the TWF-2 function is on the effects on spermathecal contractility, the only presented experimental evidence is the effects on embryonic lethality and deformation, which are considered indirect effects. However, the spermatheca exhibits complex contractility when embryos are transported, and it is not clear how *twf-2* knockout affects this process (contraction or relaxation?). The expression pattern of TWF-2 in Fig. 1 also suggests that TWF-2 is expressed in the spermatheca-uterine valve, which is also an important tissue to allow entry of the embryos to the uterus and may be affected by *twf-2* knockout.

We agree that embryonic lethality and deformation are indirect effects of changes in spermathecal contractility and that the spermatheca and its valves exhibit complex contractility during oocyte/embryo transits. To address this, we recorded with DIC microscopy ovulation movies of control, *spv-1(ok1498)*, *twf-2(msn210)* and double mutant *spv-1(ok1498);twf-2(msn210)* worms (new Figure 6A) and quantified transit times through each valve and residency time in the spermatheca (valve-to-valve time; new Figure 6B). This analysis revealed that in *spv-1* mutants, oocytes transit through the distal neck for a significantly longer time than wild type, presumably because the excessive contractility of the spermatheca bag resists the incoming oocyte. Importantly, loss of *twf-2* rescued the long transit time through the distal valve (Figure 6D). Since oocyte pinching in *spv-1* worms often occurs while they are entering the spermatheca, reducing the entry time will result in fewer such events and this is how the *twf-2* deletion rescued embryo morphology and embryonic lethality in *spv-1* worms. Transit through the sp-ut valve and valve-to-valve times were also affected in *spv-1* mutants; however, *twf-2* loss did not rescue them.

2. Overall, lack of detectable phenotypes in TWF-2 knockout alone and very modest phenotypic modification in CAP-1 RNAi and SPV-1 knockout suggest that TWF-2 as a regulator of contractility is an overstatement.

The phenotypic modification in *cap-1 RNAi* and *spv-1* knockout, while not a complete rescue, is statistically significant and biologically relevant. While *cap-1 RNAi* led to 47±19% embryonic lethality in wild-type worms, the lethality dropped by 40% (to 28±10%) in *twf-2* null worms. Similarly, loss of *twf-2* led to a 43% reduction in embryonic lethality of *spv-1* mutant (from 49.96±14% lethality in *spv-1* mutant to 28.4±11% in the double *spv-1;twf-2* mutant).

To further support the involvement of TWF-2 in actomyosin contractility, we performed new experiments in which we phalloidin stained extruded spermathecae from wild-type and *twf-2* null worms that were treated with *cap-1 RNAi* (new Figure 4F,G). Similar to the results with *spv-1* (Figure 7), we found that the increase in F-actin observed following *cap-1 RNAi* is rescued in the absence of *twf-2*.

Taken together, we show that although *twf-2* null worms do not have a detectable phenotype on their own, loss of *twf-2* rescues the increase in F-actin observed in both *spv-1* mutant and *cap-1*(RNAi), which rescues spermathecal hypercontractility, which rescues oocyte transit time through the distal neck of the spermatheca, which rescues embryo morphology, which rescues embryonic lethality.

Nevertheless, to avoid overstating the role of TWF-2, we replaced the word “regulates” with the word “modulates” in the title, abstract, introduction, and discussion, and every place where we connect between TWF-2 and contractility.

3. Although SPV-1 is specifically expressed in the spermatheca, CAP-1 is expressed in embryos. The expression pattern of TWF-2 in Fig. 1 suggests that TWF-2 is not expressed in oocytes and embryos. Is this true? Otherwise, TWF-2 and CAP-1 may be functionally related in embryonic development. Effects of *twf-2* knockout on the F-actin organization in the spermatheca of *cap-1* is not examined, and how *twf-2* and *cap-1* function together in this tissue is unclear.

As shown in a new supplementary Figure 3B, TWF-2 is expressed in embryos, but not in oocytes. We do not rule out a potential functional interaction between TWF-2 and CAP-1 during embryonic development, but such a relationship remains beyond the scope of this study, which is focused on somatic tissues, and in particular the spermatheca.

As mentioned in response to the previous comment, we added a new experiment to address spermathecal F-actin in *cap-1 RNAi* in control and *twf-2* worms, as requested by the reviewer. Phalloidin staining of extruded spermatheca and their quantification are shown in new Figure 4F,G. The results show an increase in F-actin levels upon *cap-1 RNAi*, which is completely rescued upon loss of *twf-2*.

4. The authors conclude that TWF-2 knockout suppresses the hypercontractility phenotype of SPV-1 knockout by reducing F-actin without affecting myosin. However, TWF-2 knockout does not affect F-actin in wild-type and indeed reduces phosphomyosin in wild-type (Fig. 5C), suggesting that TWF-2 may affect myosin. Then, what is the interpretation for function of TWF-2 in wild-type background? Also, TWF-2 induces actin depolymerization and removes capping protein from F-actin in vitro (Fig. 3). Therefore, TWF-2 knockout is expected to increase F-actin and stabilize the barbed ends, which is opposite to the in vivo results in the SPV-1 knockout (Fig. 5B). These data that do not align with the conclusion are not discussed well and potentially lead to an alternative conclusions.

Our antibody staining data indicate that *twf-2* loss reduces the hypercontractility phenotype of *spv-1* knockout by reducing F-actin, without affecting myosin. While *twf-2* knockout alone shows decreased phosphomyosin levels, this change does not translate into a detectable contractility phenotype as observed by ovulation dwell times as well as embryo morphologies and lethality rates, suggesting functional compensation under wild-type conditions. This explanation has been added to the discussion section. Our interpretation for the main function of TWF-2 in wild-type is to promote F-actin elongation by uncapping CP. TWF-2 is a negative regulator of CP and CP stops F-actin from elongating, producing short F-actin filaments. Hence, the presence of TWF-2 favours longer actin filaments.

Therefore, we don't agree with the reviewer's prediction that "TWF-2 knockout is expected to increase F-actin and stabilize the barbed ends". Contrarily, our prediction, based on its function, is that TWF-2 loss will decrease F-actin, which is exactly what we see in *cap-1 RNAi* and *spv-1* null conditions. We therefore don't see any contradictory results to discuss.

5. The observations that the cortical localization of TWF-2 depends on UNC-70 or SPC-1. These are potentially interesting, but what is the relevance to cell contractility in this work?

A key aim of our study was to characterise the *in vivo* function of TWF-2, which includes characterisation of its subcellular localisation and the mechanism for its recruitment. Our observation that TWF-2 cortical localisation depends on spectrins SPC-1 and UNC-70 is a part of this characterisation. It implies that cortical spectrin is essential for TWF-2's positioning at the basolateral membrane, where it can then interact with the actin cytoskeleton and its regulators. The relevance to contractility is in two ways: Firstly, as established in prior work (Wirshing & Cram, 2018), spectrin SPC-1 itself is required for production and maintenance of actin bundles in the spermatheca to regulate contractility. Secondly, as per our results, by recruiting actin regulators like TWF-2 to the cortex, the spectrin cytoskeleton may play a broader role in fine-tuning the actin dynamics and thus regulating contractility.

Minor points

1. Second paragraph of Results. The pharynx is considered a muscle tissue.

Our characterisation of pharynx as a contractile tissue acknowledges both its muscular properties and its unique organisation features. The pharynx is indeed a contractile organ composed of seven cell types, including muscle cells. While it functions similarly to muscle tissue, its complex cellular composition and developmental origin distinguish it from somatic muscles.

2. Table S1B. cdap-2 is duplicated.

Thanks for pointing the error. We have corrected TableS1B.

3. The ColP proteomics data should be disclosed by depositing to a public database.

We have submitted the proteomic data to the PRIDE database of EMBL-EBI.

Referee cross-comments

I respect the difference in opinions from Reviewer 2. My concerns are not changed. However, I do agree that live imaging of spermatheca contractility would provide more direct evidence for the involvement of TWF-2 in contractility.

Reviewer #1 (Significance (Required)):

Twinfilin has been studied extensively in vitro, but only in limited model organisms in vivo. The major strengths of this study is the use of a twinfilin-knockout mutant in *C. elegans* and quantitative phenotypic analysis. The in vitro experiments are excellent, but mostly confirms what are already known for the role of twinfilin. The major weaknesses are the lack of detectable phenotypes of the twinfilin knockout worms (when no other mutation/depletion is combined) and very modest phenotypic modifications when capping protein was depleted or a RhoGAP is mutated. Embryonic lethality was interpreted as a consequences of abnormal spermathecal contractility, but the lethality could be caused by other reasons, and the experimental evidence is considered weak. On the other hand, this study makes it clear that twinfilin is not a major regulator of contractility. Therefore, it is not a high-impact study appealing to a broad audience but it is a solid work to report phenotypes and expression patterns of twinfilin in *C. elegans* for limited audience working in the related field.

Reviewer #2

Summary:

The spermatheca must be robust to multiple rounds of stretch and contraction as eggs pass through. The authors show TWF-2 is expressed in contractile tissues, including the spermatheca, where it localizes to the cortex in a spectrin-dependent manner. They show that TWF-2 accelerates depolymeration of actin filaments. They show that loss of twf-2 can partially suppress the embryonic lethality caused by loss of capping protein (CAP-1) and a Rho GAP (SPV-1). This paper shows how twinfillin plays a role in tissue contractility, presumably through modulating F-actin dynamics.

Major comments:

The claims and conclusions are well supported by the data. However, I recommend making it more clear in the abstract that deletion of twf-2 does not cause a phenotype on its own.

Thank you for the comments. We added to the abstract the sentence: "In vivo, loss of TWF-2 alone does not produce an obvious phenotype, but..."

Fig. 4 H and I it is not immediately apparent how loss of twf-2 is suppressing the spv-1 emb phenotype given these data. I see that there are fewer points in the lower right quadrant. Is this because spv-1 mutants 'munch' the embryos and the double mutants don't? Perhaps it would help to describe more fully the interpretation of this data in the text.

In Fig. 4H and I, the key difference is in the reduction in embryos exhibiting extreme morphologies (lower left quadrant) in the *spv-1;twf-2* double mutants, compared to the *spv-1* single mutant. This reflects fewer embryos being “munched” by the spermathecal neck, which is a direct consequence of *twf-2* loss rescuing the hypercontractile spermatheca of *spv-1* mutant. This interpretation is further supported by our analysis of ovulation movies (Figure 6), which shows that *twf-2* loss rescued the prolonged oocyte transit time at the spermathecal neck in *spv-1* mutants. We expanded the text to make this connection clearer: “The hypercontractility of *spv-1* mutant spermathecae manifests itself in abnormal embryonic morphology due to abnormal squeezing and pinching of the embryos while they transit through the spermatheca (Tan & Zaidel-Bar, 2015). Morphometric analysis revealed that *twf-2* loss significantly restored embryo shape in *spv-1* mutants, with embryo area and axial ratio distributions returning toward wild-type values (Figure 5D, F-I).”

In Fig. 5, are you certain about the specificity of this pMyoII staining? It looks a bit generally speckly. It would help to include a *mel-11* or myosin light chain knockdown control.

We are confident about the specificity of the pMyoII staining as it has been rigorously validated in our lab’s prior work quantifying phosphorylated myosin in the rachis under both control and *cap-1(RNAi)* conditions (Ray et al., 2023). Additionally, this antibody’s specificity has been well-documented in another previous publication in *C. elegans* embryos (Jenkins et al., 2006).

Generally, the data and methods are presented in a way that can be reproduced. The image analysis section should be revised to be more specific about the methods used.

The image analysis section has been revised and expanded with more specific details included for each analysis.

Minor comments:

The description of the statistical analysis (that GraphPad was used) is not adequate. Please generally describe the statistical tests used including trials factor correction methods in the methods section. The figure legends do contain this information, but the methods should have a summary.

Statistical analysis details have been added in the Methods section.

I recommend pointing out the reasons you focused on the spermathecal cells for studying subcellular localization of TWf-2 (e.g. the cells are large, exhibit apical basal polarity, exhibit dynamic actin re-arrangements and are contractile).

Reasons for focussing on the spermatheca have been added to the introduction: “The spermatheca, which contains a highly organized actomyosin network that drives ~150 rounds of ovulation, provides an ideal model to dissect how actin regulators control tissue contractility.”

For completeness, it would be nice to see the spectrin localization in the *twf-2* crisper deletion strain (optional).

We crossed the *twf-2* knockout strain with SPC-1::mKate strain and the results can be found in the Supplementary Figure 2E. Similar to *twf-2* RNAi, complete loss of *twf-2* does not affect the localisation of SPC-1.

In the image analysis section, clarify how the fluorescence intensity was normalized.

With reference to the analysis of fluorescence intensity line profiles in figure 2D-H, the term

'normalized' was not the best term to use and we now explain that "The local maxima of one fluorophore peak was identified using Find Peaks FIJI plugin and aligned across different images manually using Microsoft Excel."

There are minor editorial issues in the materials and methods, for example, inconsistent spacing between number and unit (should be one space) and some typos including the use of 'u' rather than the 'mu' symbol for micro.

Thank you for pointing out. These editorial issues have been corrected.

Please indicate what magnification was used for the egg measurements (e.g. 60X).

The magnification (100X) used to image embryos has been specified.

All legends, check spacing of n {greater than or equal to} x designations. They vary, but most of them need spaces. Also sometimes n is uppercase, sometimes lowercase, and sometimes the N is missing.

Thank you for pointing out. It has been corrected.

In all figures and in text, check spacing between gene and allele designation (e.g. spv-1(ok1498)) There should not be a space.

Thank you for pointing out. It has been corrected.

Fig. 3 indicate significance on B and C.

Statistical significance on Fig. 3B and C has been indicated.

In the Supplementary Figure 2 legend, twf-2 should be in italics. 'Null' should not be italicized.

The naming has been corrected.

In the Supplementary Figure 3 legend, the word mutant should not be italicized, and there should be a space between wild-type and (N2).

The naming has been corrected.

Referee cross-commenting

I agree with some of the concerns of Reviewer #1, however, suggest the authors might be able to address the 'regulator of contractility' claim by modifying the way the results are presented and discussed, rather than undertake a significant series of experiments. SPV-1 is a major regulator of RHO-1 and spermathecal contractility, so the genetic interaction between TWF-2 and SPV-1 does suggest a role in regulation of spermathecal contractility. The reduction in embryonic lethality (loss of the small round eggs, specifically) does suggest TWF-2 regulates contractility in the spermatheca. To more directly show this, the authors could use video imaging to look at spermathecal contractility, as the egg is entering or exiting. Loss of SPV-1 causes a hypercontractile spermathecal neck and sp-ut, these phenotypes could be relieved by loss of TWF-2. I also disagree that the reduction in cap-1 RNAi embryonic lethality is 'modest'. It looks pretty significant to me (Fig. 2A), and I'm kind of surprised twf-2 can suppress cap-1 at all.

Reviewer #2 (Significance (Required)):**Significance**

- General assessment: The major strength of this study is that it is an in vivo analysis of twinfilin function, showing an important role in contractility that impacts both embryogenesis and reproductive system function in the adult.
- With any study, there is always more that could be done, for example, there are a large number of additional actin-interacting proteins that could be assayed for genetic interactions, or perhaps it makes sense to expand to spectrin interactors. However, this is a nice, clearly presented study showing a role for twinfilin in vivo. While it is surprising that the twinfilin null does not have a phenotype on its own, the genetic interactions make it clear it does have a supporting role.
- Advance: The interaction between spectrin and twinfilin is an important new finding. It is also interesting that the *C. elegans* twinfilin is a particularly potent uncapper of F-actin.
- Audience: This article should be of broad interest in the cytoskeletal (F-actin, spectrin) fields.
- My area of expertise is quite similar to those of the authors of this article.

Reviewer #3**Summary:**

Twinfilin is a conserved cytoskeletal regulator, which promotes actin filament uncapping, sustains barbed end depolymerization and sequesters actin monomers. Here, Saini et al. confirm that also *C. elegans* twinfilin (TWF-2) uncaps actin filaments in vitro, similarly to previously shown for mammalian twinfilins. Through elegant genetic studies they reveal that interplay between TWF-2 and Capping protein controls F-actin levels in certain *C. elegans* tissues, including spermatheca, and hence regulates their contractility. Finally, they provide evidence that spectrins are critical for cortical localization of TWF-2 in spermatheca.

The data presented in the manuscript appear of very good technical quality and convincing, and the study extends our knowledge on the physiological functions of twinfilins. However, there are few minor points that should be addressed to strengthen the manuscript.

Minor comments:

1. The authors state in the 'Introduction' that twinfilins accelerate the depolymerization of actin filaments. This is not entirely accurate, because as shown initially by Hakala et al., 2021 and Shekhar et al., 2021 (for some reason the former paper is not cited here, and the latter publication is cited as Shekhar et al., 2020), twinfilin actually inhibits barbed end depolymerization of ADP-actin filaments. However, it can indeed increase depolymerization of ADP-Pi barbed ends, but because twinfilin binds ADP-Pi filament barbed ends with relatively modest affinity and also does not efficiently uncap those, it is unclear if the ADP-Pi barbed end depolymerization is relevant in cells. Perhaps most importantly, both Hakala et al., 2021 and Shekhar et al., 2021 publications demonstrated that twinfilin can sustain actin filament barbed end depolymerization under assembly promoting conditions. Thus, the authors should be more precise here to avoid any confusion.

We apologize for the typo in the year of Shekhar, et al manuscript. We have now inserted the following text to clarify the distinct effects of twinfilin on barbed ends of aged and newly assembled filaments, as well as in assembly promoting conditions: "...twinfilin is now known to interact with filament barbed ends, in a context dependent manner. It inhibits barbed-end depolymerization of ADP-actin filaments (Hakala et al., 2021), but accelerates depolymerization of ADP-Pi-actin filaments (Shekhar et al., 2021). Importantly, twinfilin-mediated barbed-end depolymerization persists even under assembly-promoting conditions in the presence of profilin-ATP-G-actin (Hakala et al., 2021; Shekhar et al., 2021)."

2. Similarly to above, the authors state in the 'Abstract' and 'Results' that TWF-2 promotes barbed end depolymerization. This again is an overstatement, because based on the data presented in Fig. 3, TWF-2 inhibits barbed end depolymerization of ADP-actin filaments by ~4-fold, and only very modestly enhances barbed end depolymerization of ADP-Pi filaments. Thus, the manuscript text should be revised accordingly.

Based on reviewer's advice, we have now rephrased the text in the abstract as follows:

"In vitro, TWF-2 inhibits depolymerization of ADP-actin filaments and modestly promotes depolymerization of ADP-Pi-actin filaments, while also rapidly displacing CAP-1 from filament barbed ends." Similar revisions were done in the results section.

3. In the 'Introduction' and 'Discussion' the authors state that while mammalian twinfilins alone accelerate filament uncapping by ~6-fold, they synergize with formin mDia1 to accelerate uncapping by over 300-fold. Here, the authors forget to mention that also V-1/myotrophin greatly accelerates the filament uncapping activity of twinfilin (Hakala et al., 2021). This should be clarified in the text.

In the introduction, we have now clarified this point by adding the following text: "Interestingly, while mammalian twinfilin alone is a relatively weak uncapper and only accelerates uncapping sixfold (Hakala et al., 2021), it synergizes with V-1/myotrophin and formin mDia1 to accelerate uncapping by over 50-fold and 300-fold, respectively (Hakala et al., 2021; Reddy et al., 2025)."

We note that we didn't find any mention of twinfilin's synergy with mDia1 in the discussion section, so are unsure what the reviewer was referring to.

4. Why is the only twinfilin in *C. elegans* named TWF-2? Could the authors briefly explain/clarify this in the 'Introduction'.

The only *C. elegans* TWF-2 has been named twinfilin-2 because humans have two twinfilins TWF1 and TWF2 and *C. elegans* TWF-2 is more similar in its sequence and therefore considered an ortholog of human TWF2. This explanation has been added to the introduction.

5. Table S1 needs more explanation. For example, why is *cdap-2* listed twice in Table S1B, and why is *cap-1* included in the table (because in the manuscript text the authors state that a homozygous *cap-1* mutant is lethal)? Thus, more text to explain what is actually shown in Table S1 A and S1 B, would be beneficial for the reader.

Cdap-1 duplication has been corrected. More explanation for tables has been added to the table in the form of an updated column: selection criteria. In this column we explain that the list of genes in A were "Identified as potential cytoskeletal interactors of TWF-2 via IP-MS" and the list of genes in B are "Known actin-interacting proteins; screened due to lack of phenotype in *twf-2* single mutant, assuming potential redundancy with TWF-2".

6. Legend to Fig.1B: 'Single mid-plane' is shown in upper panels (not left) and the 'maximum z-projection' in lower panels (not right).

Legend for Fig. 1B has been corrected.

7. Fig. 3D: Why does the capped filament elongate from its barbed end in the schematic (TWF-2 flow-in)?

Thank you for pointing this out. We believe this mistake crept in during the figure editing. We have

now corrected this.

Reviewer #3 (Significance (Required)):

This study advances our understanding on the physiological functions of twinfilin, and will hence be of interest to those studying actin dynamics, as well as to scientists interested in *C. elegans* biology. The biochemical findings on the effects of *C. elegans* twinfilin on actin filament dynamics and uncapping are largely confirmatory, because they show that *C. elegans* twinfilin has similar activities to those reported earlier for mammalian twinfilins. Nevertheless, it was important to demonstrate that these activities are conserved in evolution. Also the interplay between *C. elegans* twinfilin and capping protein in vivo is in line with earlier work on yeast and mammalian cells, but the authors have used an elegant set of genetic tools to study this. The most novel finding of the manuscript is the interplay between twinfilin and spectrins, but the underlying mechanism and its possible relevance in other organisms remains to be determined in the future. Thus, the study presents a valuable contribution to the actin dynamics field.

Specific expertise of the reviewer: Actin cytoskeleton biology and biochemistry.

Decision letter

MS ID#: dev.205265

MS TITLE: Twinfilin modulates tissue contractility through capping protein uncapping in *C. elegans*

AUTHORS: Anupreet Saini; Shir Kreizman; Ekram Towsif; Jonathan Martinez-Lopez; Iska Maimon Zielonka; Anat Nitzan; Lee Rudnik; Shashank Shekhar; Ronen Zaidel-Bar

Dear Dr Zaidel-Bar,

Thank you for sending your manuscript to Development through Review Commons.

I am happy to tell you that your manuscript has been accepted for publication in Development, pending our standard publication integrity checks.

Comments from the Reviewers:

Reviewer 1: COMMENTS ON TEXT

The authors have addressed all of my comments.

COMMENTS ON DISPLAY ITEMS

I have no comment on these items.